# T$^2$SQNet: A Recognition Model for Manipulating Partially Observed Transparent Tableware Objects

**Young Hun Kim**[*,1]   **Seungyeon Kim**[*,1]   **Yonghyeon Lee**[2]   **Frank Chongwoo Park**[1]

[1]Seoul National University, [2]Korea Institute For Advanced Study

{yhun, ksy}@robotics.snu.ac.kr, ylee@kias.re.kr, fcp@snu.ac.kr

**Abstract:** Recognizing and manipulating transparent tableware from partial view RGB image observations is made challenging by the difficulty in obtaining reliable depth measurements of transparent objects. In this paper we present the *Transparent Tableware SuperQuadric Network* (T$^2$SQNet), a neural network model that leverages a family of newly extended deformable superquadrics to produce low-dimensional, instance-wise and accurate 3D geometric representations of transparent objects from partial views. As a byproduct and contribution of independent interest, we also present *TablewareNet*, a publicly available toolset of seven parametrized shapes based on our extended deformable superquadrics, that can be used to generate new datasets of tableware objects of diverse shapes and sizes. Experiments with T$^2$SQNet trained with TablewareNet show that T$^2$SQNet outperforms existing methods in recognizing transparent objects and can be effectively used in robotic applications like decluttering and target retrieval. The code is available at https://github.com/seungyeon-k/T2SQNet-public.

**Keywords:** Transparent objects, Shape recognition, Object manipulation

## 1  Introduction

Recognizing and manipulating *transparent tableware objects* is made challenging by the difficulty in obtaining reliable depth measurements of the transparent objects [1]. The task is made even more difficult when such transparent objects are placed on shelves or against walls, offering only partial views of the object. This paper addresses the recognition of transparent objects from partial view RGB images. Specifically, we emphasize the need for a *good representation for manipulation*, one that accounts for object 3D geometries and instances. The first criterion is vital to avoid collisions and ensure effective grasping, while the second is necessary for target-driven manipulation.

Recently, methods using Neural Radiance Fields (NeRF) [2], which attempt to fit 3D field models to multi-view RGB images via differentiable rendering without using depth images, have been introduced for recognizing and grasping transparent objects [3, 4, 5]. However, since NeRF assumes the availability of all-around views, when only partial views are available, the performance of these methods may not be guaranteed.

*Learning from data* is essential to infer complete geometries from partial view information (e.g., [6] and [7]). Given the high cost of real-world datasets, simulations are often used, emphasizing the need for models robust to the sim-to-real gap. However, [6] produces representations that lack instance information and shows limited robustness to the sim-to-real gap. [7] generates instance-wise 3D representations, but these rely on depth images and are thus unsuitable for transparent objects.

In this paper, we present a novel deep learning framework that tackles all the aforementioned challenges: (i) object transparency, (ii) partial views, (iii) achieving good representations for manipulation, and (iv) bridging the sim-to-real gap. Our main contribution is twofold. Firstly, we extend

---

[*]Equal contribution

8th Conference on Robot Learning (CoRL 2024), Munich, Germany.

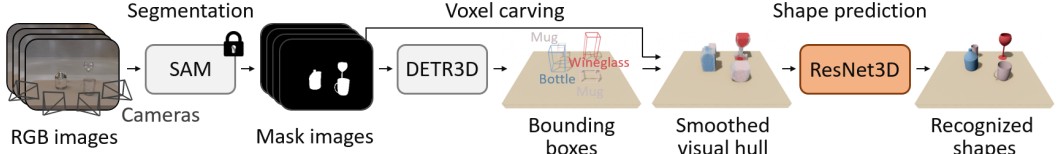

Figure 1: Model architecture of T$^2$SQNet, including: (1) a pretrained SAM for predicting object mask images [8]; (2) DETR3D for detecting object bounding boxes [9]; (3) a voxel carving module [10]; and (4) a tableware parameter prediction model based on ResNet3D [11].

the deformable superquadrics, a family of geometric shape primitives recently introduced in [7], by incorporating shearing and superparaboloids. This extension enables us to represent a wider range of tableware objects, as illustrated in Figure 2. By combining these primitives, we can define shape templates for tableware that meet the criteria for good representations in manipulation tasks.

The second contribution is the *Transparent Tableware SuperQuadric Network (T$^2$SQNet)*, a neural network model that infers the geometries of tableware objects by representing each object as a union of the extended deformable superquadrics. Our model comprises four key steps: prediction of 2D masks, 3D bounding boxes, visual hulls, and superquadric parameters (illustrated in Figure 1). There are two novel elements to our approach. First, we leverage SAM mask images [8], which is critical in enhancing robustness to the sim-to-real gap. Secondly, we devise representations based on a set of visual hulls – one for each object – as inputs to the shape prediction network. We have observed that this approach significantly improves performance compared to models that directly estimate superquadric parameters in an end-to-end manner.

As a second contribution obtained as a byproduct of our approach, but also of independent interest, we introduce *TablewareNet*, a family of continuously parameterized shapes representing everyday tableware objects across seven classes, using extended deformable superquadrics and their combinations as shown in Figure 2. Users can easily generate a new dataset with diverse sizes and shapes within each class; even a new class of objects can be constructed using our tools.

Our experiments demonstrate that T$^2$SQNet trained with TablewareNet outperforms existing state-of-the-art methods in predicting the geometries of transparent objects. This superior performance extends beyond the test set from TablewareNet to include unseen data from the large-scale transparent object dataset named TRansPose [12]. Furthermore, we show the practical applicability of our method in two downstream tasks including sequential decluttering and object rearrangement for target retrieval, highlighting the advantages of our representations.

## 2 TablewareNet: Dataset for Cluttered Transparent Tableware

In this section, we first extend the deformable superquadrics introduced in [7, 13] to represent non-convex shapes such as bowls and wineglasses. We then propose a *TablewareNet*, a family of continuously parametrized shapes representing tableware objects, where each shape is defined as a proper combination of our extended deformable superquadrics.

### 2.1 Extended Deformable Superquadrics

*Superquadrics*, parametrized by only a few parameters, can represent a relatively wide range of geometric shapes. We employ two kinds of superquadrics: *superellipsoids*, which have been used for object manipulation [14, 15, 16, 7, 17, 18, 19], and *superparaboloids*, which are newly introduced. Superellipsoids and superparaboloids are implicit surfaces with the following implicit functions with size parameters $(a_1, a_2, a_3) \in \mathbb{R}^3_+$ and shape parameters $(e_1, e_2) \in \mathbb{R}^2_+$: for $\mathbf{x} = (x, y, z)$,

$$\overbrace{f_{se}(\mathbf{x}) = \left(\left|\frac{x}{a_1}\right|^{\frac{2}{e_2}} + \left|\frac{y}{a_2}\right|^{\frac{2}{e_2}}\right)^{\frac{e_2}{e_1}} + \left|\frac{z}{a_3}\right|^{\frac{2}{e_1}} = 1}^{\text{Superellipsoid}}, \overbrace{f_{sp}(\mathbf{x}) = \left(\left|\frac{x}{a_1}\right|^{\frac{2}{e_2}} + \left|\frac{y}{a_2}\right|^{\frac{2}{e_2}}\right)^{\frac{e_2}{e_1}} - \left(\frac{z}{a_3}\right) = 1}^{\text{Superparaboloid}}.$$

By adjusting the parameters, various surfaces can be represented as shown in the upper of Figure 2. For the superparaboloid, only the region where $z \leq 0$ is used. Deformable superquadrics extend

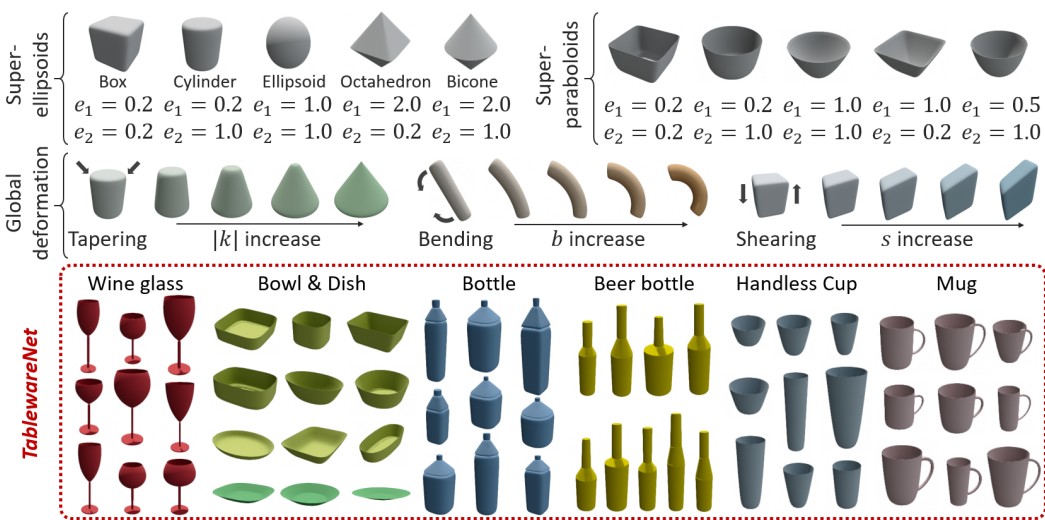

Figure 2: *Upper*: Examples of superquadrics, including superellipsoids and superparaboloids, and deformable superquadrics. *Lower*: *TablewareNet* objects as unions of deformable superquadrics.

superquadrics by incorporating global deformations. Existing works, such as [7], have employed traditional deformations including tapering and bending. *Tapering* is a transformation where the shape gradually narrows or widens along a certain direction. *Bending* is a transformation where an axis turns into a circular section. Additionally, we employ a *shearing* transformation, where an object is skewed along a specified axis. Each transformation used in this paper is expressed in the following formula (deformations are applied in the z-axis direction):

$$\mathbf{x} = \begin{bmatrix} x \\ y \\ z \end{bmatrix} \longmapsto \overbrace{D_t(\mathbf{x}) = \begin{bmatrix} t(z)x \\ t(z)y \\ z \end{bmatrix}}^{\text{Tapering}}, \quad \overbrace{D_b(\mathbf{x}) = \begin{bmatrix} x + (R-r)\cos\alpha \\ x + (R-r)\sin\alpha \\ (b^{-1}-r)\sin\gamma \end{bmatrix}}^{\text{Bending}}, \quad \overbrace{D_s(\mathbf{x}) = \begin{bmatrix} x \\ y \\ z + sx \end{bmatrix}}^{\text{Shearing}}.$$

Here, motivated by [7], we use a linear tapering function $t(z) = t_k(z) := \frac{k}{a_3}z + 1$ with a single parameter $k$. For bending, $b > 0$ and $\alpha$ represents the degree and the direction of bending in the xy plane, and other values are computed from these parameters; $r = \cos(\alpha - \text{atan2}(y, x))\sqrt{x^2 + y^2}$, $R = b^{-1} - (b^{-1} - r)\cos\gamma$, $\gamma = zb$. In shearing, $s$ leverages the intensity of shearing. These deformations are concatenated in an obvious way, i.e., $D = D_s \circ D_b \circ D_t$; we note that the deformations are not commutative. The deformable superquadric function is then given by $f_D = f \circ D^{-1}$, where $D$ and $f$ are a concatenated deformation and a superquadric, respectively. We can position the implicit function using an object pose $T \in SE(3)$ with the following expression: $f_D(T^{-1}\mathbf{x}) = 1$.

## 2.2 Tableware Templates and Synthetic Dataset Generation

**Tableware Templates.** We combine deformable superquadrics to define templates representing seven types of tableware: *wine glasses, bottles, beer bottles, bowls, dishes, handleless cups*, and *mugs*; see the lower of Figure 2. Let $\mathcal{T}$ be a tableware object constructed using $n$ deformable superquadrics $\{S_i | i = 1, \ldots, n\}$. Each superquadric $S_i$ includes function type $f_i \in \{f_{se}, f_{sp}\}$, pose $T_i \in \text{SE}(3)$, size $\mathbf{a}_i \in \mathbb{R}^3_+$, shape parameters $\mathbf{e}_i \in \mathbb{R}^2_+$ and deformation parameters $\mathbf{d}_i = (k_i, b_i, \alpha_i, s_i)$. For $\mathcal{T}$ to represent an appropriate shape, constraints among these parameters are necessary. For example, to represent an appropriate wineglass, constraints on the positions of the head and the base, as well as the length of the pillar, are needed. To easily assign these constraints, we re-parametrize the above parameters to low-dimensional *tableware parameters*; we note that there is an one-to-one mapping between tableware parameters and the superquadric parameters $f_i, T_i, \mathbf{a}_i, \mathbf{e}_i, \mathbf{d}_i$ for $i = 1, \ldots, n$. More details are found in Appendix B.1.

**Synthetic Dataset Generation.** We then generate the *TablewareNet* dataset as shown in Figure 3. By adjusting tableware parameters, we can create diverse 3D tableware meshes. Spawning these meshes in a user-defined environment (e.g., table or shelf) within PyBullet [20], a physics simulator, allows us to generate cluttered scenes. Using Blender [21], a photorealistic renderer, with transpar-

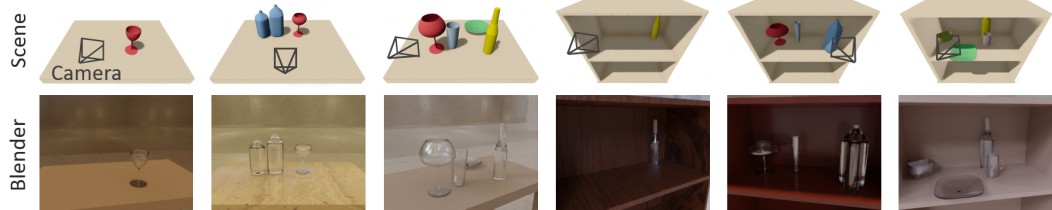

Figure 3: Examples of the *TablewareNet* dataset, including 3D geometries of transparent objects on a table or shelf (*Upper*) and RGB images rendered from Blender (*Lower*).

ent textures, we obtain RGB images of the scenes from arbitrary poses. For data generation, we first randomly spawn one to four tableware objects. Each dataset includes the poses, tableware parameters, bounding boxes of the objects, RGB images, mask images, and depth images from seven camera views. Detailed statistics can be found in Appendix B.2. Note that this dataset can be used not only to train our model but also for other data-driven recognition models. Additionally, users can create customized datasets by modifying the compositions of superquadrics.

## 3   T$^2$SQNet: Transparent Tableware SuperQuadric Network

In this section, we propose a novel framework for recognizing the 3D shapes and locations of transparent objects using only partial multi-view RGB observations. Overall, our method consists of four steps: (1) prediction of masks in 2D images, (2) prediction of 3D bounding boxes, (3) computation of a *smoothed visual hull* through voxel carving, and (4) prediction of tableware parameters (i.e., a set of parameters of the superquadrics); see Figure 1. We sequentially apply these modules during inference, which can accumulate prediction errors. Therefore, we develop techniques to train each module accurately and robustly against noise and sim-to-real gaps, which we explain in detail below.

**Mask Prediction.** Simulated RGB images are often non-photorealistic, leading to significant sim-to-real discrepancies. To mitigate this, we use the pretrained segmentation network SAM [8], trained on real images, to generate masks for the next module, the bounding box estimator. Specifically, we use SAM with a "tableware" prompt to generate masks. However, SAM sometimes incorrectly predicts parts of tableware objects as background. To address this, we apply random color-jittering five times to an image, obtain five corresponding SAM output mask images, and compute their union to derive the final mask. Empirically, this approach significantly improves mask predictions.

**3D Bounding Box Prediction.** The mask images are then input into a 3D bounding box prediction model, the DETR3D [9]. This model outputs the class probability along with the central coordinates and sizes of the bounding boxes along the $x$, $y$, and $z$ axes for tableware objects. We apply a random cut-out augmentation to a randomly selected mask image to prevent the model from relying solely on information from a specific image among the multi-view mask images. This encourages the model to extract information evenly from the multi-view masks for predictions of bounding boxes.

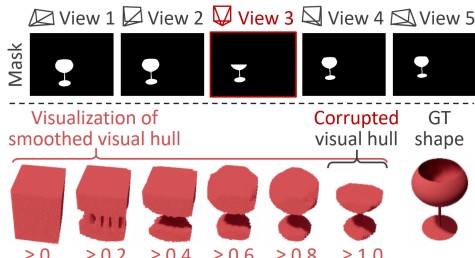

Figure 4: Visualization of the corrupted visual hull and smoothed visual hull when one mask image (view 3) is corrupted.

**Smoothed Visual Hull.** Voxel carving constructs a visual hull by removing voxels from a volumetric grid that do not match object silhouettes captured from multiple camera views [10]. Voxel carving is applied to a volumetric grid within each detected bounding box using the predicted masks as object silhouettes. We found that, although rarely, the predicted masks can confuse objects with the background, leading to significant distortion in the resulting visual hull when even one mask fails to identify object pixels and removes corresponding voxels, as shown in Figure 4. To construct more robust 3D representations, we count the number of images whose mask pixels include the projection of each corresponding 3D voxel instead of storing binary values. We then store normalized values representing the frequency of voxel appearances, which we refer to as the *smoothed visual hull*.

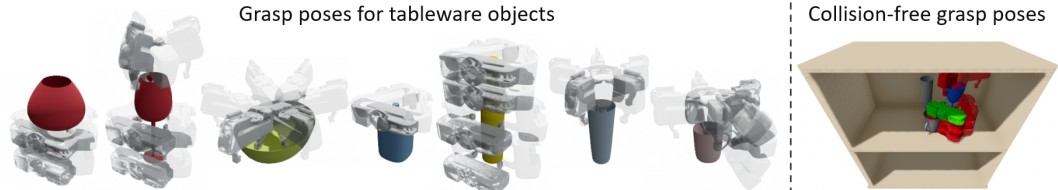

Grasp poses for tableware objects | Collision-free grasp poses

Figure 5: *Left*: 6-DoF grasp sampling results for various tableware objects. *Right*: Collision-free grasp poses for a target object $\mathcal{A}$ (blue wine glass); green grippers indicate collision-free grasp poses, while red grippers indicate poses with collisions.

**Tableware Parameter Prediction.** The smoothed visual hulls are input into neural networks based on ResNet3D [11] to predict the tableware parameter. Since each tableware class has different voxel resolutions and a varying number of shape parameters, separate predictors are assigned to each class. After our bounding box predictor outputs the position, size, and class of a bounding box, the parameter predictor for the corresponding class takes as input the smoothed visual hull carved within the predicted bounding box. To train these networks, we use the Chamfer loss between the ground truth point cloud and point cloud sampled from the predicted superquadrics in a differentiable manner. Lastly, to enhance the model's robustness to bounding box prediction errors, we apply random perturbations to the location and size of the bounding box to generate a perturbed smoothed visual hull, which is then used for robust training. Further implementation details on our overall methods can be found in Appendix B.4.

## 4 Geometry-Aware Object Manipulation with T²SQNet

T²SQNet offers many practical advantages for downstream object manipulation tasks. It enables faster recognition compared to approaches that require real-time optimization, such as NeRF variants [3, 4]. Moreover, its instance-wise object recognition facilitates target-driven manipulation.

A distinctive advantage of T²SQNet arises from its use of extended deformable superquadrics for output representations. First of all, the implicit function representations for their surfaces enable rapid collision checks. Suppose the surface meshes of the robot's links and gripper are represented by surface points. We can determine if the robot is in collision with the recognized objects by assessing whether the implicit function values of the surface points are greater than or equal to zero.

Additionally, we can easily design an effective 6-DoF grasp sampler (see the left of Figure 5). Given the superquadric representations, we can intuitively design grasp pose distributions, which adapt continuously (even differentiably) almost everywhere as the superquadric parameters change. As demonstrated with the wine glass examples in the left of Figure 5 (the top part of the wine glass on the far left can be grasped, whereas that of the other wine glass cannot), the distributions may not change continuously at some shapes. Nonetheless, for the most part, the distributions change differentiably, offering the potential for applying gradient-based grasp pose modulation.

We conclude this section with our graspability-aware object rearrangement algorithm, which demonstrates the effective use of the grasp pose sampler and rapid collision checking in downstream robotic tasks. Consider a scenario in which a target object $\mathcal{A}$ is initially not graspable, surrounded by a set of objects. Our objective is to determine how to reconfigure the surrounding objects through pick-and-place actions to make $\mathcal{A}$ graspable. To achieve this, we explore the action space to identify an action or a sequence of actions that result in $\mathcal{A}$ becoming graspable. This process requires rapid grasp pose sampling and computation of the graspability of $\mathcal{A}$, which we accomplish by assessing collisions between grippers at sampled grasp poses and the surrounding objects; see the right of Figure 5. We refer to further details including grasp sampler and object rearrangement to Appendix B.5.

## 5 Experiments

In this section, we empirically demonstrate that (i) our proposed model, T²SQNet, surpasses existing state-of-the-art models in recognizing transparent objects, and (ii) our method is highly effective in various robotic applications, including sequential decluttering and target retrieval tasks.

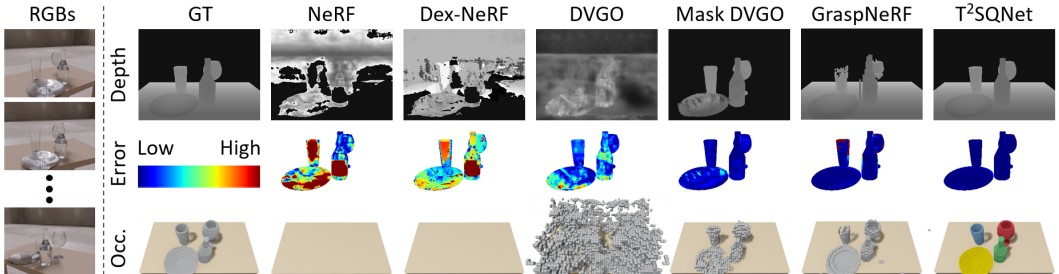

| RGBs | | GT | NeRF | Dex-NeRF | DVGO | Mask DVGO | GraspNeRF | T²SQNet |

Figure 6: Recognition results from RGB images from a test set of *Tableware* dataset. *Upper*: Reconstructed depth images, *Middle*: Depth reconstruction errors within object masks, *Lower*: Reconstructed occupancy maps; for T²SQNet, the occupancy grid is colored by the instance.

**Environment.** We use the 7-DoF Franka Emika Panda robot equipped with a parallel-jaw gripper and an Intel RealSense D435 camera mounted on the gripper. The raw visual input consists of a sequence of seven RGB images captured from different robot poses. Figure 7 illustrates the real-world robot environment, including the robot, camera, table, shelf, and the transparent objects used in the experiments.

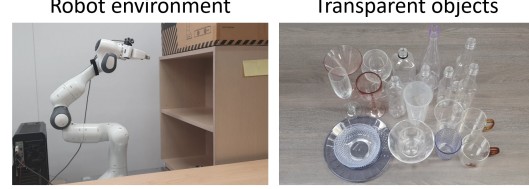

Figure 7: Real-world robot environment setting and various transparent objects.

**Baseline Methods.** We compare T²SQNet with the following baseline methods: *NeRF* [2], *Dex-NeRF* [3], *DVGO* [22], DVGO using segmentation masks (denoted as *Mask DVGO*) adopted from [5], and *GraspNeRF* [6]. The differentiable rendering-based methods, including NeRF, Dex-NeRF, DVGO, and Mask DVGO, input multi-view RGB images to fit a 3D opacity field, which is then pre-processed into other representations (e.g., depth image or occupancy map) as needed. For the segmentation mask for Mask DVGO, we use the SAM [8]. GraspNeRF is a generalizable NeRF-based model that inputs multiple RGB images from random view-poses and predicts a voxelized truncated signed distance field (TSDF). Detailed implementations can be found in Appendix C.1.

### 5.1 Transparent Object Recognition Performance

In this section, we demonstrate the performance advantages of T²SQNet over existing methods. We adopt two metrics to evaluate 3D recognition quality: *depth accuracy* and *volumetric occupancy IoU*. Depth accuracy is the ratio of predicted depth values that fall within a certain threshold of the ground-truth depth values in a depth image. This accuracy is measured only for pixels where the ground-truth object is present, using depth images obtained from the view-poses used during training. Volumetric occupancy IoU is is the ratio of the intersection to the union between ground-truth occupancy voxel grid and predicted occupancy voxel grid. Further details for calculating evaluation metrics for each method can be found in Appendix C.1.

We compare the recognition performances of T²SQNet, trained with TablewareNet, and the baseline methods using (i) 40 scenes from the test set of TablewareNet, with 10 scenes each containing one to four objects, and (ii) 40 scenes from the TRansPose dataset [12], with 10 scenes each containing one to four tableware objects. Figure 6 shows the recognition results, including reconstructed depth images (with depth reconstruction error maps) and occupancy grids on a test set of TablewareNet. T²SQNet exhibits the lowest depth error and demonstrates the best performance in terms of 3D occupancy, accurately delivering instance information.

Table 1 shows the quantitative recognition results. Differentiable rendering-based methods struggle due to sparse partial views. Specifically, the opacity field seems to concentrate near the camera lens, resulting in lower depth accuracy in Dex-NeRF, which measures depth based on the distance to an opacity exceeding a certain threshold, compared to NeRF. Among these methods, Mask DVGO has the best performance but still shows relatively low performance from an occupancy IoU perspective. GraspNeRF achieves higher volumetric IoU performance than other differentiable rendering-based methods, likely due to its use of supervised learning to train on ground truth TSDF values. How-

Table 1: Depth and occupancy reconstruction accuracy on TablewareNet and TRansPose dataset.

| | TablewareNet | | | | TRansPose [12] | | | |
| | Depth | | | Occ. | Depth | | | Occ. |
| METHOD | $\delta_{0.05}$ | $\delta_{0.10}$ | $\delta_{0.20}$ | IoU | $\delta_{0.05}$ | $\delta_{0.10}$ | $\delta_{0.20}$ | IoU |
|---|---|---|---|---|---|---|---|---|
| NeRF [2] | 0.068 | 0.134 | 0.310 | 0.000 | 0.062 | 0.122 | 0.279 | 0.000 |
| Dex-NeRF [3] | 0.061 | 0.115 | 0.278 | 0.000 | 0.018 | 0.037 | 0.131 | 0.000 |
| DVGO [22] | 0.144 | 0.288 | 0.664 | 0.029 | 0.158 | 0.307 | 0.646 | 0.017 |
| Mask DVGO [5] | 0.697 | 0.885 | 0.963 | 0.394 | 0.785 | 0.928 | **0.962** | 0.487 |
| GraspNeRF [6] | 0.771 | 0.827 | 0.888 | 0.554 | 0.716 | 0.778 | 0.881 | 0.551 |
| T$^2$SQNet (ours) | **0.944** | **0.958** | **0.968** | **0.740** | **0.918** | **0.936** | 0.961 | **0.636** |

ever, GraspNeRF faces challenges in training due to its high-dimensional voxel output representation compared to T$^2$SQNet. T$^2$SQNet, with its lower-dimensional output, benefits from easier training and better generalization performance, resulting in the best overall performance. Additionally, both GraspNeRF and T$^2$SQNet show competitive results on the TRansPose dataset, indicating that TablewareNet includes a sufficiently diverse and reliable set of tableware shapes. Further experimental results with more example figures are provided in Appendix D.1.

## 5.2 Object Manipulation Performance

In this section, we demonstrate the effectiveness of our model, T$^2$SQNet, on two object manipulation tasks: (i) *sequential declutter*, which involves sequential grasping in a cluttered environment, and (ii) *target retrieval*, which involves object rearrangement planning to retrieve an initially non-graspable target object[1]. The target object is indicated by a target tableware class name (e.g., wineglass).

**Sequential Declutter.** For sequential declutter, T$^2$SQNet first recognizes the geometries of the tableware objects from input RGB images, as shown on the left of Figure 8. Grasp poses are then generated for each object using a 6-DoF grasp sampler, as described in Chapter 4. Among the generated grasp poses, those for which inverse kinematics are solved are selected, and the grasp pose with the least likelihood of collision is chosen, as detailed in Chapter 4. After grasping and removing an object, the next grasp pose is generated to execute sequential grasping. Figure 8.

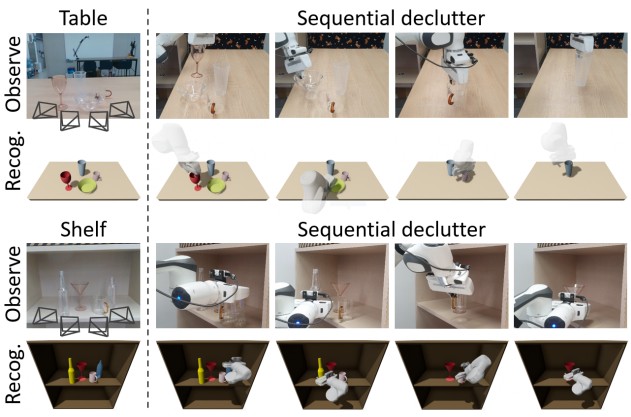

Figure 8: Examples of sequential declutter experiment results on a table and a shelf.

shows an example of sequential declutter experiment results. Based on the accurately predicted geometries of the objects, our method succeeds in sequentially grasping the objects without re-recognition, while avoiding collisions with other objects and the environment.

We compare the sequential declutter performances of T$^2$SQNet, *Mask DVGO* (which shows the best recognition performance among differentiable rendering-based methods), and *GraspNeRF*. We use the depth image-based pretrained FC-GQ-CNN [23] and the TSDF-based Volumetric Grasping Network (VGN) [24] for grasp pose generation in Mask DVGO and GraspNeRF, re-

Table 2: Real-world declutter success rates in single (S) and cluttered (C) environments for table and shelf.

| | Shelf | | Table | |
| METHOD | S | C | S | C |
|---|---|---|---|---|
| Mask DVGO [5] | 1/5 | 0/5 | 0/5 | 0/5 |
| GraspNeRF [6] | 3/5 | 1/5 | 1/5 | 0/5 |
| T$^2$SQNet (ours) | **5/5** | **4/5** | **4/5** | **3/5** |

spectively. The experiments are conducted five times for each scenario: one object (single) or four objects (cluttered) within the scene, for both table and shelf environments. Success is defined as the robot successfully sequentially grasping all objects in the scene. The experimental details for

---

[1]The real-world manipulation videos can be found at https://youtu.be/6m5ZOrbSxxI.

sequential clutter can be found in Appendix C.2. Table 2 shows the sequential declutter success rates in real-world experiments and demonstrates that our method outperforms the other baseline methods by significant margins. Failure of baselines mainly results from inaccurate recognition results. Further experimental results and discussions are provided in Appendix D.2.

**Target Retrieval.** For target retrieval, we first define a *graspability function*, which is set to 1 if the object is graspable and 0 otherwise. Given a target object, we sample grasp poses for the recognized target and check for collisions of the grippers with the environment (e.g., shelf) and surrounding other objects. The graspability function value is then set to 1 if at least one collision-free grasp pose exists, and 0 otherwise. Further implementation details are provided in Appendix B.4.

Figure 9 shows an example of target retrieval experiment results. $T^2$SQNet recognizes the tableware objects and checks whether there is exactly one object of the target class; in this case, the target object is a wineglass. To make the target object graspable, we sample some pick-and-place actions and select action sequences that maximize the graspa-

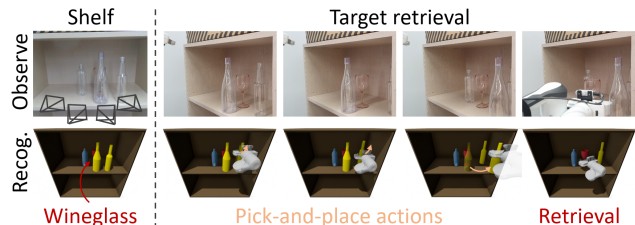

Figure 9: Examples of target retrieval experiment results on a shelf (the target object is the wineglass).

bility function. The experimental details can be found in Appendix C.3. $T^2$SQNet-based method successfully rearranges the surrounding objects and finally retrieves the target wineglass. Note that since we currently use a sampling-based method and a discrete reward function, it is not optimal in terms of the number of actions (i.e., unnecessary rearranging actions appear), but this can be further improved with advanced object rearrangement methods [25, 26, 27, 18]. Our method achieves an 80% success rates in the five trials. More details are provided in Appendix D.3.

## 5.3 Limitations and Future Directions

First, $T^2$SQNet currently cannot recognize objects from unknown classes because it relies on known shape templates. One future direction is to exploit research that attempts to represent complex objects as multiple superquadrics without predefined templates [28, 7, 29, 30]. Second, we need to use a fixed set of camera view poses for both training and testing because DETR3D, which we use to detect bounding boxes of the objects, requires initially fixed camera view poses. To overcome this limitation, we need to develop a model or learning technique capable of handling various camera poses, like the method used in GraspNeRF that aggregates 2D image features in 3D volume space to make the model capable of arbitrary views [6]. Lastly, when multiple objects extensively overlap in images, the performance of 3D bounding box estimation is likely to decline since masks alone may not sufficiently reveal object boundaries. Utilizing instance segmentation information is thus an interesting direction for future work.

## 6 Conclusion

We have introduced a novel model, *$T^2$SQNet* and a novel dataset, *TablewareNet*, designed to recognize and manipulate transparent tableware objects from partial-view RGB images. By leveraging deformable superquadrics and TablewareNet dataset, $T^2$SQNet achieves accurate, instance-wise geometric representations essential for effective manipulation tasks. Our experiments demonstrate that $T^2$SQNet outperforms existing state-of-the-art methods in predicting the geometries of transparent objects and excels in sequential declutter and target retrieval tasks.

**Acknowledgments**

Y. H. Kim, S. Kim, and F. C. Park were supported in part by IITP-MSIT grant RS-2021-II212068 (SNU AI Innovation Hub), IITP-MSIT grant 2022-220480, RS-2022-II220480 (Training and Inference Methods for Goal Oriented AI Agents), IITP-MSIT grant RS-2024-00436680 (Collaborative Research Projects with Microsoft Research) under the Global Research Support Program in the Digital Field program, KIAT grant P0020536 (HRD Program for Industrial Innovation), SRRC NRF grant RS-2023-00208052, SNU-AIIS, SNU-IPAI, SNU-IAMD, SNU BK21+ Program in Mechanical Engineering, SNU Institute for Engineering Research, and Microsoft Research Asia. Y. Lee was the beneficiary of an individual grant from CAINS supported by a KIAS Individual Grant (AP092701) via the Center for AI and Natural Sciences at Korea Institute for Advanced Study.

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

# Appendix

# A    Related Works

## A.1    NeRF-based Transparent Objects Recognition for Manipulation

NeRF [2] and its variants use differentiable rendering to find optimal 3D vision models (e.g., 3D color fields, occupancy fields) from multiple 2D RGB images. Due to NeRF's ability to learn 3D structures without using depth images, several studies [3, 4] have attempted to use NeRF for grasping transparent objects. Dex-NeRF [3] has first proposed transparent object recognition and grasping using NeRF, introducing a depth rendering technique suitable for grasping differing from traditional NeRF approaches. This work utilizes the rendered depth images to grasp objects using a pre-trained Dex-Net [31]. Building on this, Evo-NeRF [4] has enhanced performance by improving the time efficiency of the training process and adding a geometry regularizer term. Recently, NFL [5] has employed pre-trained mask estimators and 2D normal field estimators to more accurately capture 3D geometry.

The primary limitation of these NeRF-based methods is their dependency on having access to all-around, hemispheric views. When only partial views are available, these methods experience catastrophic failures. Furthermore, while NeRFs effectively capture the overall geometry of a scene, they do not provide details about specific object instances. This absence of instance-specific information poses significant challenges for tasks requiring target-driven manipulation.

## A.2    Learning-based Transparent Objects Recognition for Manipulation

In this section, we summarize learning-based methods developed for transparent object recognition and manipulation. One of the dominant approaches attempts to refine a corrupted depth image using information from an RGB image, and then apply existing depth image-based grasp pose generation algorithms [32]. For instance, Cleargrasp [1] has proposed a method that takes an RGB-D image of a scene containing transparent objects, uses the RGB portion to estimate surface normals, detect boundaries, and segment objects with a neural network, and then globally optimizes these outputs to restore the damaged depth image. DepthGrasp [33] has developed a GAN-based generator to directly produce a completed depth image from raw RGB-D input. Similarly, TransCG [34] has provided a dataset of raw and refined depth images of real-world transparent objects and train a U-Net structure neural network to predict refined depth images from RGB images and incomplete depth images. Moreover, SwinDRNet [35] has proposed depth image completion using a two-stream Swin Transformer [36], introducing domain randomization to tackle sim-to-real domain shift issues.

However, the grasp pose generation methods based on depth images have fundamental limitations. First, full 6-DoF grasping is not possible, resulting in reduced diversity of grasp poses. Second, there is often a lack of information on the 3D geometry of objects, making it difficult to generate collision-free grasping trajectories. Third, most cases do not involve instance-wise representation, which complicates target-driven manipulation.

GraspNeRF [6] predicts TSDF values from multiple RGB images and trains a model to predict grasp poses using VGN [24]. This method overcomes some of the disadvantages mentioned above. Since it directly generates grasp poses, it can also be trained to create 6-DoF grasp poses. Additionally, because it outputs TSDF values, it has information on 3D geometry, enabling the generation of collision-free trajectories. However, since the current version does not provide instance information, target-aware manipulation is challenging. One of the most significant differences in our research is that we use extended deformable superquadrics for 3D scene representation, which, compared to TSDF, is more memory-efficient and allows for much faster collision checks and grasp sampling.

We would like to emphasize that all the learning-based methods mentioned above face a sim-to-real issue. Because RGB images in simulation differ from real images, models trained on simulated RGB images commonly experience a significant performance drop when applied to real images. It would be ideal to use real-world data, but collecting it is quite challenging. Although various methods

have been introduced in previous studies [35, 6] to enhance robustness to the sim-to-real gap, none have been particularly effective. In our research, we developed a model that uses mask images as input, which we found to be significantly more robust to the sim-to-real gap compared to models that directly take RGB images as input.

Additionally, [37] combines rendering-based techniques for calculating light refraction and reflection on transparent surfaces with learning methods for normal estimation and shape refinement. Such research excels in providing highly detailed surface representations of objects without shape prior when (i) the environment map is fully known, (ii) the object's center position is known, and (iii) all-around views are available from the real-world. Meanwhile, our method, while offering relatively constrained surface expressiveness using extended deformable superquadrics, can predict the shape of objects in unknown poses within unknown environments, even with only partial views.

# B Implementation Details for Our Methods

## B.1 Details for TablewareNet Objects

In this section, we describe the constraints on the superquadric parameters and poses for each tableware object mentioned in Section 2.2. – wine glasses, bottles, beer bottles, bowls, dishes, handleless cups, and mugs – and explain how these constraints are represented using tableware parameters.

**Notations.** Following the notation used in Section 2.2., each tableware object is composed of $n$ superquadrics $\{S_i | i = 1, \ldots, n\}$, with their function type, pose, size parameter, and shape parameter denoted as $f_i, T_i, \mathbf{a}_i$, and $\mathbf{e}_i$, respectively. Additionally, their deformation parameters are denoted as $\mathbf{d}_i = (k_i, b_i, \alpha_i, s_i)$. If there is no deformation, $\mathbf{d}_i = (0, 0, 0, 0)$; we denote the absence of bending as $b_i = 0$ for convenience, as bending diminishes when $b$ approaches zero. In actual implementation, deformation $D_b$ is not computed when $b_i = 0$.

**Wine Glass.** The wine glass template encompasses a wide range of wine glass and champagne glass shapes. This template consists of three superquadrics: two superellipsoids $S_1$, $S_2$ for the foot and stem, and a superparaboloid $S_3$ for the bowl. The tableware parameters and the constraints on the superquadrics are given in Figure 10 and Table 3.

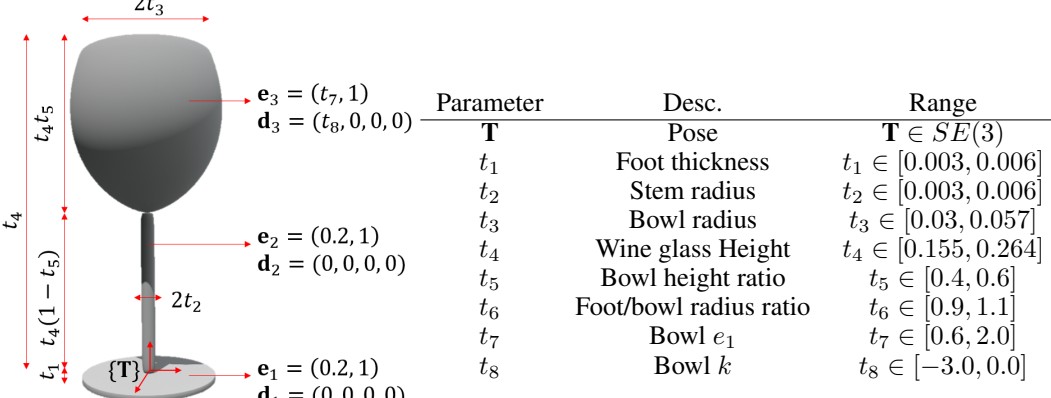

| Parameter | Desc. | Range |
|---|---|---|
| **T** | Pose | $\mathbf{T} \in SE(3)$ |
| $t_1$ | Foot thickness | $t_1 \in [0.003, 0.006]$ |
| $t_2$ | Stem radius | $t_2 \in [0.003, 0.006]$ |
| $t_3$ | Bowl radius | $t_3 \in [0.03, 0.057]$ |
| $t_4$ | Wine glass Height | $t_4 \in [0.155, 0.264]$ |
| $t_5$ | Bowl height ratio | $t_5 \in [0.4, 0.6]$ |
| $t_6$ | Foot/bowl radius ratio | $t_6 \in [0.9, 1.1]$ |
| $t_7$ | Bowl $e_1$ | $t_7 \in [0.6, 2.0]$ |
| $t_8$ | Bowl $k$ | $t_8 \in [-3.0, 0.0]$ |

Table 3: Tableware parameters of wine glasses.

Figure 10: Description for the wine glass parameters.

**Bowl.** The bowl template consists of one superquadric: a superparaboloid $S_1$ for the bowl. The tableware parameters are given in Figure 11 and Table 4.

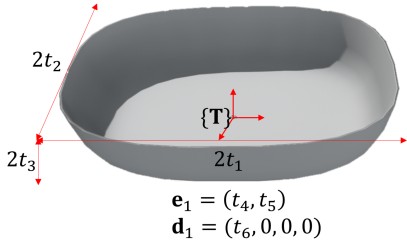

$\mathbf{e}_1 = (t_4, t_5)$
$\mathbf{d}_1 = (t_6, 0, 0, 0)$

| Parameter | Desc. | Range |
|---|---|---|
| **T** | Pose | $\mathbf{T} \in SE(3)$ |
| $t_1$ | Bowl $a_1$ | $t_1 \in [0.04, 0.15]$ |
| $t_2$ | Bowl $a_2$ | $t_2 \in [0.04, 0.15]$ |
| $t_3$ | Bowl $a_3$ | $t_3 \in [0.02, 0.1]$ |
| $t_4$ | Bowl $e_1$ | $t_4 \in [0.01, 0.3]$ |
| $t_5$ | Bowl $e_2$ | $t_5 \in [0.1, 1.0]$ |
| $t_6$ | Bowl $k$ | $t_6 \in [-0.1, 0.3]$ |

Figure 11: Description for the bowl parameters.

Table 4: Tableware parameters of bowls.

**Bottle.** The bottle template consists of three superquadrics: superellipsoids $S_1, S_2, S_3$ for the body, shoulder and finish. The tableware parameters and the constraints on the superquadrics are given in Figure 12 and Table 5.

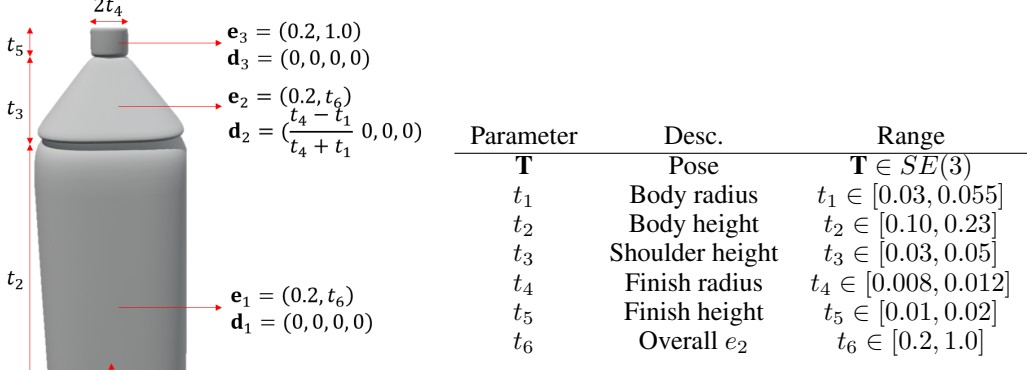

$\mathbf{e}_3 = (0.2, 1.0)$
$\mathbf{d}_3 = (0, 0, 0, 0)$

$\mathbf{e}_2 = (0.2, t_6)$
$\mathbf{d}_2 = (\frac{t_4 - t_1}{t_4 + t_1}, 0, 0, 0)$

$\mathbf{e}_1 = (0.2, t_6)$
$\mathbf{d}_1 = (0, 0, 0, 0)$

| Parameter | Desc. | Range |
|---|---|---|
| $\mathbf{T}$ | Pose | $\mathbf{T} \in SE(3)$ |
| $t_1$ | Body radius | $t_1 \in [0.03, 0.055]$ |
| $t_2$ | Body height | $t_2 \in [0.10, 0.23]$ |
| $t_3$ | Shoulder height | $t_3 \in [0.03, 0.05]$ |
| $t_4$ | Finish radius | $t_4 \in [0.008, 0.012]$ |
| $t_5$ | Finish height | $t_5 \in [0.01, 0.02]$ |
| $t_6$ | Overall $e_2$ | $t_6 \in [0.2, 1.0]$ |

Table 5: Tableware parameters of bottles.

Figure 12: Description for the bottle parameters.

**Beer bottle.** The beer bottle template consists of three superquadrics: superellipsoids $S_1, S_2, S_3$ for the body, shoulder and neck. The tableware parameters and the constraints on the superquadrics are given in Figure 13 and Table 6.

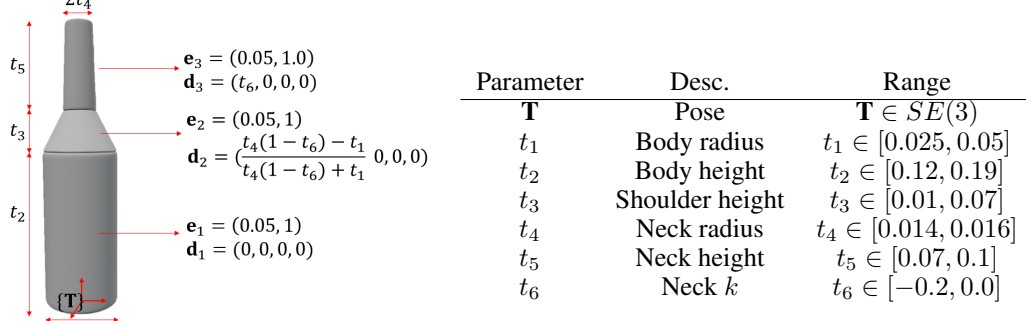

$\mathbf{e}_3 = (0.05, 1.0)$
$\mathbf{d}_3 = (t_6, 0, 0, 0)$

$\mathbf{e}_2 = (0.05, 1)$
$\mathbf{d}_2 = (\frac{t_4(1 - t_6) - t_1}{t_4(1 - t_6) + t_1}, 0, 0, 0)$

$\mathbf{e}_1 = (0.05, 1)$
$\mathbf{d}_1 = (0, 0, 0, 0)$

| Parameter | Desc. | Range |
|---|---|---|
| $\mathbf{T}$ | Pose | $\mathbf{T} \in SE(3)$ |
| $t_1$ | Body radius | $t_1 \in [0.025, 0.05]$ |
| $t_2$ | Body height | $t_2 \in [0.12, 0.19]$ |
| $t_3$ | Shoulder height | $t_3 \in [0.01, 0.07]$ |
| $t_4$ | Neck radius | $t_4 \in [0.014, 0.016]$ |
| $t_5$ | Neck height | $t_5 \in [0.07, 0.1]$ |
| $t_6$ | Neck $k$ | $t_6 \in [-0.2, 0.0]$ |

Table 6: Tableware parameters of beer bottles.

Figure 13: Description for the beer bottle parameters.

**Handless Cup.** The handless cup template consists of one superquadric: a superparaboloid $S_1$. The tableware parameters are given in Figure 14 and Table 7.

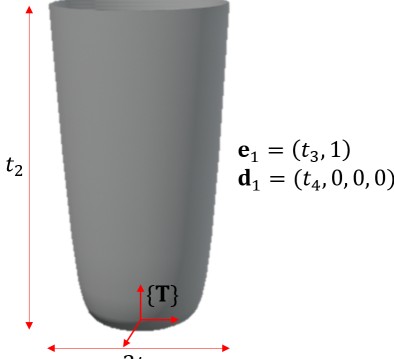

$e_1 = (t_3, 1)$
$d_1 = (t_4, 0, 0, 0)$

| Parameter | Desc. | Range |
|---|---|---|
| **T** | Pose | $\mathbf{T} \in SE(3)$ |
| $t_1$ | Cup radius | $t_1 \in [0.025, 0.05]$ |
| $t_2$ | Cup height | $t_2 \in [0.05, 0.22]$ |
| $t_3$ | Cup $e_1$ | $t_3 \in [0.01, 0.3]$ |
| $t_4$ | Cup $k$ | $t_4 \in [0.0, 0.3]$ |

Table 7: Tableware parameters of handless cups.

Figure 14: Description for the handless cup parameters.

**Mug.** The mug template consists of two superquadric: superparaboloids $S_1$, $S_2$ for the cup and handle. The tableware parameters and the constraints on the superquadrics are given in Figure 15 and Table 8.

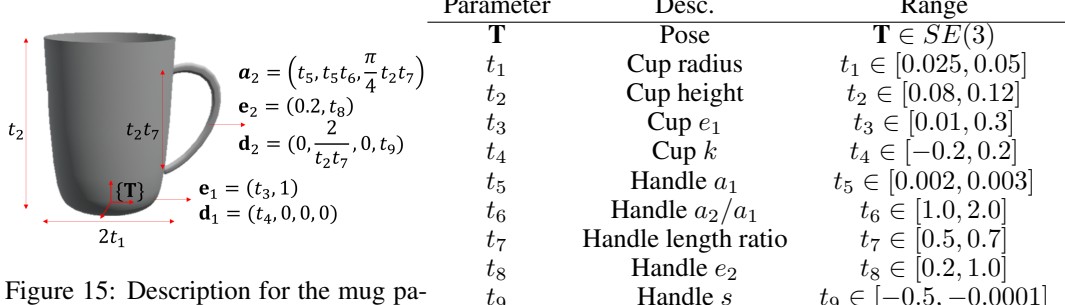

$a_2 = \left(t_5, t_5 t_6, \frac{\pi}{4} t_2 t_7\right)$
$e_2 = (0.2, t_8)$
$d_2 = \left(0, \frac{2}{t_2 t_7}, 0, t_9\right)$
$e_1 = (t_3, 1)$
$d_1 = (t_4, 0, 0, 0)$

| Parameter | Desc. | Range |
|---|---|---|
| **T** | Pose | $\mathbf{T} \in SE(3)$ |
| $t_1$ | Cup radius | $t_1 \in [0.025, 0.05]$ |
| $t_2$ | Cup height | $t_2 \in [0.08, 0.12]$ |
| $t_3$ | Cup $e_1$ | $t_3 \in [0.01, 0.3]$ |
| $t_4$ | Cup $k$ | $t_4 \in [-0.2, 0.2]$ |
| $t_5$ | Handle $a_1$ | $t_5 \in [0.002, 0.003]$ |
| $t_6$ | Handle $a_2/a_1$ | $t_6 \in [1.0, 2.0]$ |
| $t_7$ | Handle length ratio | $t_7 \in [0.5, 0.7]$ |
| $t_8$ | Handle $e_2$ | $t_8 \in [0.2, 1.0]$ |
| $t_9$ | Handle $s$ | $t_9 \in [-0.5, -0.0001]$ |

Figure 15: Description for the mug parameters.

Table 8: Tableware parameters of mugs.

**Dish.** The dish template consists of one superquadric: a superparaboloid $S_1$. The tableware parameters are given in Figure 16 and Table 9.

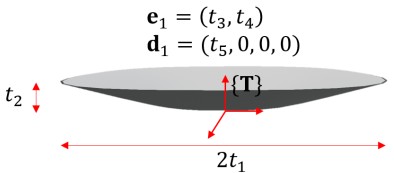

$e_1 = (t_3, t_4)$
$d_1 = (t_5, 0, 0, 0)$

| Parameter | Desc. | Range |
|---|---|---|
| **T** | Pose | $\mathbf{T} \in SE(3)$ |
| $t_1$ | Dish radius | $t_1 \in [0.08, 0.14]$ |
| $t_2$ | Dish height | $t_2 \in [0.015, 0.03]$ |
| $t_3$ | Dish $e_1$ | $t_3 \in [0.01, 0.3]$ |
| $t_4$ | Dish $e_2$ | $t_4 \in [0.5, 1.0]$ |
| $t_5$ | Dish $k$ | $t_5 \in [0.3, 0.6]$ |

Figure 16: Description for the dish parameters.

Table 9: Tableware parameters of dishes.

## B.2 Details for Tableware Dataset

Our dataset employs uniform random sampling for each object class, offering infinitely continuous variations in object shapes, unlike other datasets with a finite number of fixed shapes. We generate TablewareNet objects using uniformly random sampled shape parameters and spawn them in the physics simulator for each scene. Since the parameters of objects in all scenes are sampled from a uniform distribution, it is rare for exactly the same object to be spawned. For each scene, we provide mask images, depth images, and RGB images from seven different camera poses using the synthetic camera parameters of the RealSense D435. Additionally, each scene includes 3D geometry information including object poses, tableware parameters, class labels, bounding boxes, meshes, and TSDF values. Currently, The dataset features two versions: one with transparent objects on a shelf and another with on a table. Each dataset includes 128,000 scenes, with 32,000 scenes each containing 1, 2, 3, or 4 objects. The training, validation, and test data consist of 120,000, 4,000, and 4,000 scenes, respectively.

**Comparison with Existing Dataset.** Here we provide brief comparisons between the Tableware Dataset and existing datasets containing transparent objects. TRans10K [38] consists of real RGB images of transparent objects with segmentation masks generated by manual annotation, but it only includes segmentation information and lacks 3D information such as pose annotations or object CAD models. Cleargrasp [1] and TODD [39] provide real-world images of transparent objects along with annotated pose information for those objects with 3D CAD models, but they include only a small number of object types, no more than 10. ClearPose [40] and TRansPose [12] offer annotation-rich, large-scale real-world datasets containing relatively many transparent objects, up to 100. However, the number of objects used in data generation is still small for recognizing a wide variety of transparent objects.

Recently, Syn-TODD [41] has provided a large-scale dataset with diverse object sets using synthetic objects and a photorealistic renderer named Blender. Our dataset is also in the spirit of Syn-TODD, as it aims to generate a dataset containing various objects using synthetic objects. The main differences between TablewareNet and Syn-TODD is the procedure of generating synthetic objects. The authors of Syn-TODD generate synthetic objects using combinations of linear, polynomial, and sinusoidal functions with various parameters. This procedural generation provides a variety of shapes but can also produce unrealistic transparent objects, and objects with multiple parts (e.g., mugs with handles) are difficult to generate. Meanwhile, we have generated transparent objects that are realistic and can have multiple parts by utilizing extended deformable superquadrics. However, due to the smoothness of the superquadric, it is difficult to express surfaces with many curvatures, such as those of a cup. We believe that the two transparent object generation methods have different advantages and disadvantages and that they can complement each other in the future.

## B.3 Examples of Class Supplementation Using Deformable Superquadrics

In this section, we present examples of additional objects that can be generated using deformable superquadrics, which are not covered in our dataset. **Perfume Bottle.** Perfume bottles come in various shapes and are transparent objects commonly seen in daily life. However, they are not included in our dataset because they are not tableware. The perfume bottle template consists of two superquadrics: superellipsoids $S_1, S_2$ for the body and cap. **Plastic Cup with Spherical Lid.** Plastic cups with spherical lids are easily seen in the most coffee shops. This template consists of two superquadrics: superellipsoids $S_1, S_2$ for the cup and lid. The example of perfume bottle and cup with spherical lid represented by deformable superquadrics can be found in Figure 17.

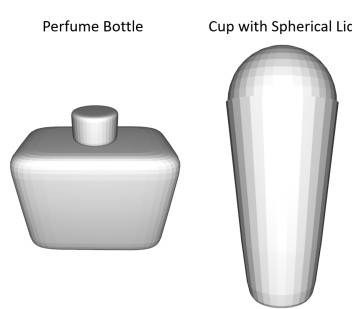

Perfume Bottle  Cup with Spherical Lid

Figure 17: Perfume bottle and plastic cup with spherical lid represented by deformable superquadrics.

### B.4 Details for Model Architecture and Training Process

#### B.4.1 Bounding Box Prediction Model Architecture

In this section, we describe the structure of the neural network used for bounding box prediction. We utilize the architecture of DETR3D [9], which takes fixed multi-view RGB images as input and outputs multiple 3D bounding boxes. DETR3D first obtains 2D features from the multi-view images using ResNet [42] and FPN [43]. It employs a Transformer architecture, where each layer decodes candidate positions for bounding box centers using a sub-network for object queries. These decoded positions are projected onto each image plane, and the corresponding image feature values are incorporated into the object queries using multi-head attention. The refined object queries are then used as input for the next layer. From each object query passing through the layers, the bounding box center, size, and class are predicted using an MLP structure for training. During inference, the bounding boxes and classes are predicted from the object queries of the final layer. For more details, refer to [9].

We adopt the DETR3D structure with a few modifications. Instead of using a classifier to predict the object class for each object query, we pre-assign classes to the object queries and use a confidence predictor to predict a value in $[0, 1]$ whether the query corresponds to a present object. For example, if there are 3500 object queries and 7 classes, the first 500 queries represent the first class, and if the confidence predictor assigns high confidence (we use a threshold value of 0.75) to one of these queries, it is predicted that a bounding box for an object of that class exists. Unlike the original DETR3D using RGB image inputs, we use binary mask images as inputs.

#### B.4.2 Bounding Box Prediction Model Training

**Matching and Loss Calculation.** As in the original DETR3D, we use bipartite matching to calculate the loss. Since we have pre-assigned classes to each query, the matching occurs within each class. We use binary cross-entropy for the confidence scores and $L_1$ loss for the bounding box coordinates and size. The weights between the binary cross-entropy loss and the $L_1$ loss are set to 5:1. After calculating the loss for the optimal matching within each class, we sum these losses across all classes to obtain the final loss value.

**Training Process.** We train the network using 12,000 scenes from the TablewareNet dataset; although our model can utilize the 120,000 scenes, we use only the 12,000 samples that include RGB images for comparison with other RGB-based baselines. We employ the Adam optimizer with an initial learning rate of 0.00005. A cosine annealing scheduler is used, and the training process spans 200 epochs. Using a validation set composed of 400 scenes, we defined the best model as the one with the highest mAP, calculated with a bounding box IoU threshold of 0.75.

**Cutout Augmentation.** For cutout augmentation, the number of holes per mask image is determined by uniformly sampling an integer between 0 and 2. The size of each hole is determined by uniformly sampling an integer between 50 and 100 for both the width and height; note that mask image size is $(240, 320)$. The position of each hole is also uniformly sampled within the image pixels.

#### B.4.3 3D Voxel Representation of Smoothed Visual Hull

We need to voxelize the predicted bounding boxes with fixed voxel size $L_i$, to create the raw 3D space to be carved. Here, $i$ represents the class index. Given that bounding boxes can vary in size, voxelizing them directly would result in different resolution of voxels, disabling the inference of the 3D CNN and FCN architecture. To address this, we follow a specific procedure to standardize the raw voxel representation.

First, we inspect the all size of the ground truth bounding boxes of class $i$ and store the maximum size $(W_{i_{\max}}, H_{i_{\max}}, D_{i_{\max}})$. Using this maximum size, we create a standardized 3D voxel space centered on the predicted bbox center. This ensures that all voxel representations have a consistent resolution $(W_{i_{\max}}/L_i, H_{i_{\max}}/L_i, D_{i_{\max}}/L_i)$.

Next, we calculate the smoothed visual hull within this standardized voxel space. To further refine the representation, we add a channel to the voxel grid, where voxels inside the predicted bounding box are assigned a value of 1, and those outside are assigned a value of 0. The final voxel input to the network thus takes the shape $(W_{i_{\max}}/L_i, H_{i_{\max}}/L_i, D_{i_{\max}}/L_i, 2)$, with the first channel representing the smoothed visual hull and the second channel indicating bounding box occupancy. This standardized approach ensures consistent and accurate input for the 3D CNN, facilitating reliable network inference.

### B.4.4  Tableware Parameter Prediction Model Architecture

We employ a ResNet3D [11] + FCN architecture to predict the tableware parameters, including the pose **T**, using the smoothed visual hull voxel as input. Given that each tableware class has different voxel resolutions and the number of parameters, we train separate predictors for each class. In other words, there are as parameter predictors as the number of the classes, with each predictor dedicated to estimating the parameters of a single class of tableware objects.

In addition, we assume that all objects are upright. Therefore, instead of predicting the entire $SE(3)$ matrix for the object pose **T**, the model only predict the relative position $p \in \mathbb{R}^3$ to the bounding box center and the rotation around the $z$-axis, $\theta \in [0, 2\pi)$.

### B.4.5  Tableware Parameter Prediction Model Training

**Chamfer Distance as Loss function.** The tableware parameters output by the parameter predictor could be directly trained via supervised learning using the $L_1$ or $L_2$ distance from the ground truth tableware parameters. However, this approach may not yield a zero loss value even when the predicted object shape matches the ground truth shape due to symmetrical ambiguities. For instance, orientations of objects with circular symmetry such as wine glasses, beer bottles, handless cups, and dishes, do not affect their shapes. In the case of a bottle, rotating the object by 90 degrees results in the same shape, thus the ground truth shape has four possible orientations. For bowls, swapping the width $t_1$ and length $t_2$ and rotating the orientation by 90 degrees along the $z$-axis results in the same shape. We want a loss function that yields a zero value when the predicted shape matches the ground truth shape, regardless of these symmetrical ambiguities.

To address this, we use the Chamfer distance between point clouds sampled from the predicted and ground truth object surfaces as the loss function. This requires a differentiable point cloud sampling method from deformable superquadric parameters, which we will discuss later. For a tableware object composed of multiple superquadrics $S_1, \ldots, S_n$, we compute the Chamfer distance for each deformable superquadric part. Specifically, if $P_{i,\text{pred}}$ and $P_{i,\text{gt}}$ are the point clouds sampled from the predicted and ground truth $i$-th deformable superquadric $S_{i,\text{pred}}$ and $S_{i,\text{gt}}$, respectively, the Chamfer loss is defined as:

$$\sum_{i=1}^{n} \text{chamfer}(P_{i,\text{pred}}, P_{i,\text{gt}})$$

**Differentiable Point Sampling on Deformable Superquadrics.** For superquadrics without deformation, there exists an explicit parameterization as follows;

$$\mathbf{x}_{se} = \begin{bmatrix} x \\ y \\ z \end{bmatrix} = \begin{bmatrix} a_1 \cos^{e_1} \theta \cos^{e_2} \phi \\ a_2 \cos^{e_1} \theta \sin^{e_2} \phi \\ a_3 \sin^{e_1} \theta \end{bmatrix}, \mathbf{x}_{sp} = \begin{bmatrix} x \\ y \\ z \end{bmatrix} = \begin{bmatrix} a_1 h \cos^{e_2} \phi \\ a_2 h \sin^{e_2} \phi \\ a_3(h^{2/e_1} - 1) \end{bmatrix},$$

where $-\pi/2 \leq \theta \leq \pi/2$, $-\pi \leq \phi \leq \pi$ and $0 \leq h \leq 1$, and $\cos^e \theta := \text{sgn}(\cos\theta)|\cos\theta|^e$ and $\sin^e \theta := \text{sgn}(\sin\theta)|\sin\theta|^e$.

We utilize a uniform grid in $\theta$, $\phi$, and $h$ coordinates to sample points on the superquadric surface using this explicit parameterization. This method is differentiable with respect to the superquadric parameters.

When dealing with deformations, we apply the transformations $D_t$, $D_b$, and $D_s$ described in Section 2.1 to these sampled points. These transformations are also differentiable with respect to deformation parameters.

**Training Process.** During training, we observe empirically that adding $L_1$ loss for tableware parameters (excluding the object's pose) and MSE loss for position as regularizers accelerate the training process. We do not use a regularizer for orientation due to symmetry issues previously discussed. The weights used between the position MSE loss, parameter $L_1$ loss, and Chamfer loss were set to $1 : 0.1 : 1$. The model is trained with 12,000 data which include RGB images. The input smoothed visual hulls are generated using ground truth bounding boxes and mask images for each scene. Similar to the bounding box predictor training, we employed the Adam optimizer with an initial learning rate of 0.00005, using a cosine annealing scheduler. The training process spanned 200 epochs. Using a validation set composed of 400 scenes, we defined the best model as the one with the lowest chamfer distance.

**Data Augmentation with Bounding Box Perturbation.** During inference, errors in the bounding box predictions may lead to low performance of the parameter predictor if it is trained solely on visual hulls generated from ground truth bounding boxes. To address this, we add noise to the bounding box's position and size to create a perturbed visual hull during training. Specifically, as mentioned in B.4.3., while generating the voxel representation, we apply perturbations to the center and size of the bounding boxes when creating the second channel representing the bounding box occupancy. This data augmentation strategy ensures that the parameter predictor remains robust to prediction errors from the bounding box predictor, enabling T²SQNet to maintain high performance even when bounding box predictions are not perfectly accurate.

## B.5 Details for Geometry-Aware Object Manipulation

**Collision Checking.** The implicit function representations for deformable superquadrics enable precise and rapid collision checks. Suppose the surface meshes of the robot's links and gripper are represented by surface points $\mathcal{P}_r \subset \mathbb{R}^3$. In practice, we represent the last two links (i.e., links 6 and 7 of the 7-DoF Franka Emika Panda robot) and the grippers of the robot, where collisions are likely to occur, as a point cloud $\mathcal{P}_r = \{\mathbf{x}_j^r \in \mathbb{R}^3\}_{j=1}^{n_r}$, where $n_r$ is the number of points of the point cloud. Let a recognized object be composed of $n$ extended deformable superquadrics $\{S_i\}_{i=1}^n$. Each deformable superquadric $S_i$ is represented by an implicit function $S_i(\mathbf{x}) = f_i \circ D_i^{-1}(T_i^{-1}\mathbf{x}) = 1$, where $f_i \in \{f_{se}, f_{sp}\}$ is the superquadric equation of the $i$'th part defined by superquadric parameters, $T_i$ is the pose of the $i$'th part, and $D_i$ is the deformation of the $i$'th part.

The main idea to check for collision is that a point $\mathbf{x} \in \mathbb{R}^3$ is inside the $i$'th part $S_i$ when the value $S_i(\mathbf{x})$ is less than 1; otherwise, it is outside. Using this fact, we can determine if the robot is in collision with the recognized objects by assessing whether the implicit function values of $S_i$ for the surface points are greater than or equal to zero. The collision can be determined by

$$\mathbb{1}(\min_{i,j} S_i(\mathbf{x}_j^r) > 1)$$

where $i = 1, ..., n$ is the object index, $j = 1, ..., n_r$ is the point index of the robot point cloud $\mathcal{P}_r$, and $\mathbb{1}(\cdot)$ is the indicator function. If the value is 1, there is no collision between the robot and the recognized object; if it is 0, there is a collision.

**6-DoF Grasp Sampler.** After obtaining the deformable superquadric representation of the transparent objects, we sample feasible grasp poses from the obtained shapes. While it is possible to find the antipodal point by utilizing the implicit function of the deformable superquadric and its closed-form normal vector [7], we manually design a faster and more diverse grasp sampler in this paper. Since each object consists of multiple deformable superquadric parts, we first generate grasp poses for each part. Inspired by previous works [17], we manually generate top-down and side grasp poses for the superellipsoids according to their shapes. For superparaboloids, we generate grasp poses that grasp the edge. Example grasp poses can be found in Section 4. After generating grasp poses for each part, we check whether the grasp poses avoid self-collision with the object. Grasp poses with

a distance between the antipodal points greater than 7.5 cm are removed from the candidates, as the maximum gripper width of the Franka gripper is 8 cm.

**Details for Sequential Declutter.** Suppose that $n_o$ objects are recognized and the $i$'th recognized object is composed of $n_i$ extended deformable superquadrics $\{S_{ij}\}_{j=1}^{n_i}$. We additionally represent the environment (e.g., table or shelf) using superquadric implicit functions and consider the environment as the $(n_o + 1)$'th object; for example, a shelf can be composed of five boxes, so it is represented by five superquadric implicit functions $\{S_{(n_o+1)j}\}_{j=1}^{n_{n_o+1}}$.

For sequential declutter, we first sample the grasp poses using the 6-DoF grasp sampler for all recognized objects. For each recognized object, we sample up to 30 grasp poses. For each grasp pose, we manually design the grasping trajectory, in which the gripper approaches about 20 cm along the z-direction of the gripper. Among the grasping trajectories, we reject those for which inverse kinematics cannot be solved. We denote each grasping trajectory by index $k$.

After generating the grasping trajectories, we should check whether the robot following the trajectory collides with the surrounding tableware objects or the environment (e.g., table or shelf). For the $k$'th grasping trajectory, we obtain an afterimage mesh of the robot and the gripper and obtain the point cloud sampled from the afterimage mesh; we denote the point cloud as $\mathcal{P}_k = \{\mathbf{x}_l^k \in \mathbb{R}^3\}_{l=1}^{n_r}$, where $n_r = 2048$. We can determine that the $k$'th grasping trajectory does not collide with the recognized objects and the environment if the value

$$\mathbb{1}(\min_i \min_{j,l} S_{ij}(\mathbf{x}_l^k) > 1)$$

is one for $i = 1, \ldots, n_o + 1$, $j = 1, \ldots, n_i$, and $l = 1, \cdots, n_r$. Sometimes there may be multiple grasping trajectories that do not result in a collision. In this case, the grasping trajectory with the largest value of $\min_i \min_{j,l} S_{ij}(\mathbf{x}_l^k)$ for $k$ is selected. Since all these computations are parallelizable, grasp planning for sequential declutter can be performed in real-time.

**Details for Target Retrieval.** Suppose that $n_o$ objects *excluding the target object* are recognized and the $i$'th recognized object is composed of $n_i$ extended deformable superquadrics $\{S_{ij}\}_{j=1}^{n_i}$. We also represent the environment using superquadric implicit functions and consider the environment as the $(n_o + 1)$'th object.

For target retrieval, we define a graspability function, which is set to 1 if the object is graspable and 0 otherwise. To compute the graspability function, we first sample $n_g$ kinematically feasible grasping trajectories for the target object. As in the case of sequential decluttering, we denote the afterimage point cloud of the robot and the gripper for the $k$'th grasping trajectory as $\mathcal{P}_k = \{\mathbf{x}_l^k \in \mathbb{R}^3\}_{l=1}^{n_r}$, where $n_r = 2048$. The target object is defined to be graspable if there is at least one collision-free grasping trajectory. Therefore, the graspability function can be calculated by:

$$\mathbb{1}(\max_k \min_i \min_{j,l} S_{ij}(\mathbf{x}_l^k) > 1),$$

for $i = 1, \ldots, n_o + 1$, $j = 1, \ldots, n_i$, $l = 1, \cdots, n_r$, and $k = 1, \cdots, n_g$. This function can also be calculated in real-time through parallel computation after recognition is performed.

To make the initially non-graspable target object graspable, we maximize the graspability function using pick-and-place actions with a sampling-based model predictive control (MPC) approach. Since the dynamics of the objects after the pick-and-place actions are precisely known, we do not use an additional learning-based model to predict the dynamics. When sampling pick-and-place actions, it is important not only to find a collision-free trajectory for grasping the object to be picked-and-placed but also to identify placeable locations for the object. Therefore, it is crucial to check whether the robot collides with the recognized objects and the environment. The collision checking using superquadric representation makes this possible precisely and rapidly as described above. Among these sampled pick-and-place actions, we find the action that maximizes the function the most, and then repeat the same process to perform pick-and-place actions until the target object becomes graspable.

# C  Experimental Details

## C.1  Additional Details for Recognition Experiments

**Baseline Implementations.** *NeRF and Dex-NeRF* are trained using the seven partial views available in TablewareNet and use the same views for depth rendering. We use the model from the final epoch of training for evaluation. We follow the approach described in [3] for depth rendering of NeRF and Dex-NeRF, with a $\sigma$ threshold of 15 for Dex-NeRF. To generate occupancy voxel grids, we render depth images from uniformly sampled view poses over a hemisphere using the trained NeRF and Dex-NeRF models. These depth images are converted to TSDFs, and regions with negative TSDF values are marked as occupied. *DVGO and Mask DVGO* are also trained using the seven partial views. For depth rendering, both methods employ the depth rendering technique used by NeRF in [3]. Predicted occupancy voxels are defined as those with an occupancy probability exceeding a threshold of 0.1, based on the trained DVGO and Mask DVGO models. For *GraspNeRF*, we train the model with the TablewareNet dataset, where the training and validation data consist of 12,000 and 100 samples, respectively. We use rendering loss and TSDF loss, where ground truth TSDF values are used to derive the occupancy voxels directly from the predicted TSDF values. In detail, GraspNeRF is trained on *TablewareNet* dataset with slight modifications: we fix the camera view poses during both training and testing to match the input view poses of $T^2$SQNet. We do not train the volumetric grasp detection module simultaneously, nor do we use the Eikonal regularization term in the geometry loss. Depth images are directly rendered from the reconstructed TSDF values. For our model, *$T^2$SQNet*, we train the model with the TablewareNet dataset, where the training and validation data consist of 120,000 and 1,000 samples, respectively. We obtain depth images and occupancy voxels using the implicit function representation of the superquadrics of the predicted tableware objects.

**Test Dataset.** The test dataset consists of (i) 40 scenes from the test set of TablewareNet, with 10 scenes each containing one to four TablewareNet objects, and (ii) 40 scenes generated from TRansPose objects, with 10 scenes each containing one to four transparent objects. For test dataset generation using TRansPose objects, we first select some transparent objects from the TRansPose dataset [12]. The TRansPose objects we use are shown in Figure 18. Using the TRansPose objects, we follow the same process as when generating the TablewareNet dataset, as described in Section 2.2. We first spawn random objects from the TRansPose objects within PyBullet and then render RGB images of the scenes using Blender with transparent textures.

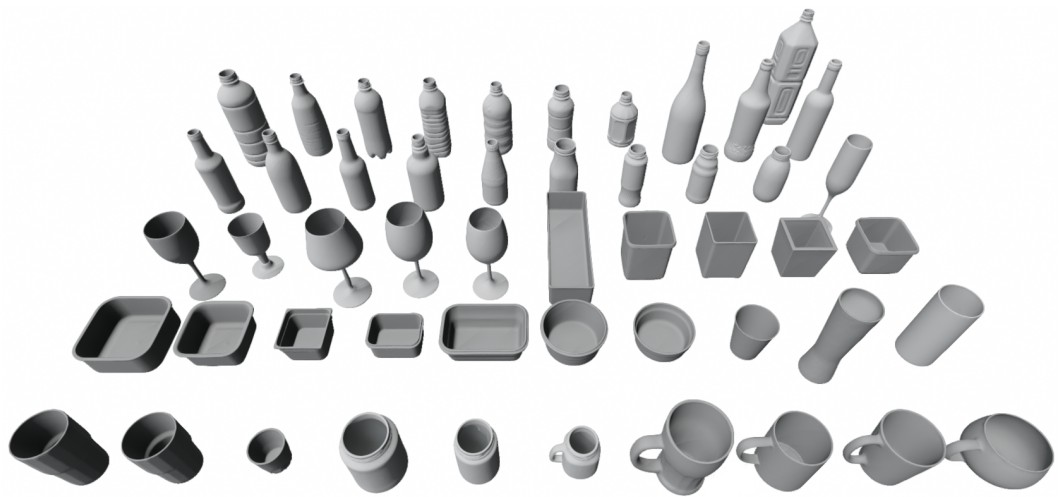

Figure 18: TRansPose objects used for test dataset [12].

**Runtime of $T^2$SQNet.** We recall that the $T^2$SQNet consists of four steps: (1) prediction of masks in 2D images, (2) prediction of 3D bounding boxes, (3) computation of a smoothed visual hull

through voxel carving, and (4) prediction of tableware parameters (i.e., a set of parameters of the superquadrics).

(1) The average inference time for one RGB image using the segmentation model (i.e., SAM) is 0.69 seconds. We use a total of 7 camera views and infer a total of 35 images by performing color jittering 5 times on each camera view. Therefore, the average time it takes to obtain the 2D mask is 24.2 seconds.

(2) It takes an average of 0.28 seconds to infer 3D bounding boxes using 7 mask images as input with DETR3D.

(3) It takes an average of 0.15 seconds to obtain a smoothed visual hull from the bounding box of each object through voxel carving.

(4) It takes an average of 0.004 seconds to predict tableware parameters via ResNet3D on the smoothed visual hull of each object.

The average total runtime of T$^2$SQNet, when there are 4 objects in a scene and it takes 7 RGB images as input to output the superquadrics of all objects, is 25 seconds. If SAM could be inferred in batches, the overall runtime could be greatly reduced. All computation times are measured with a GeForce RTX 3090 and a Gen Intel(R) Core(TM) i9-11900K @ 3.50GHz.

## C.2 Additional Details for Sequential Declutter Experiment

We generate five scenarios for each number of objects, one (single) and four (cluttered), and for each environment type, including shelf and table, resulting in a total of 20 scenarios. The initial settings for all scenes, including the configuration and poses of objects, can be seen in Figure 19. When testing the three baselines, including Mask DVGO, GraspNeRF, and T$^2$SQNet, we place the same objects in the same pose as consistently as possible for each scene.

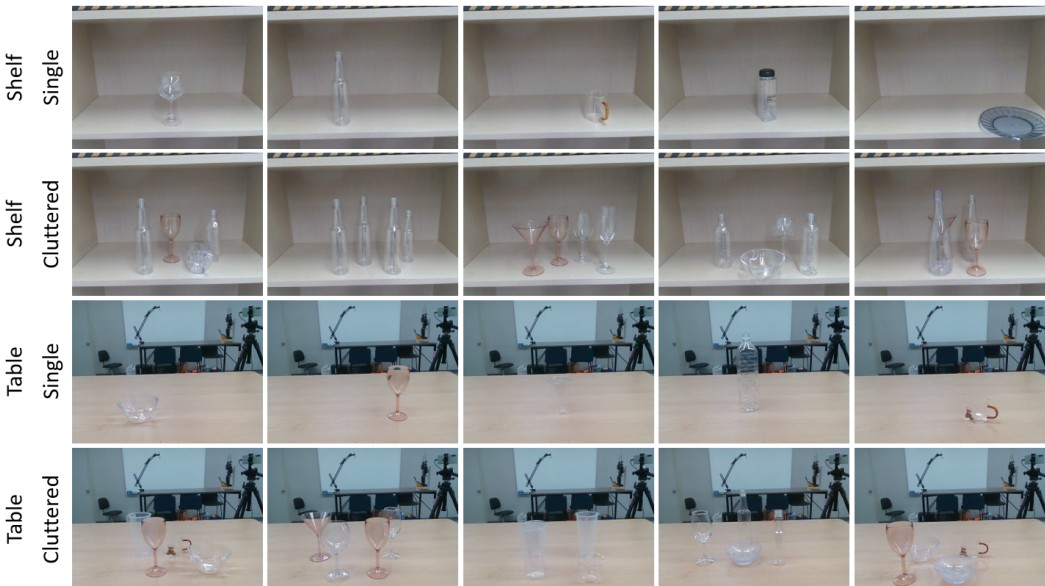

Figure 19: Initial scene settings for real-world sequential declutter experiments.

## C.3 Additional Details for Target Retrieval Experiment

We generate five scenarios with designated target objects in a shelf environment. The initial scene settings, including the target tableware class name (e.g., wine glass), for all scenarios can be found in Figure 20. For the target retrieval experiment, we place the target object so that it is not initially graspable.

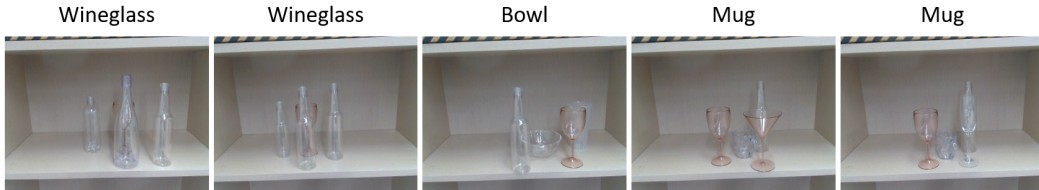

Figure 20: Initial scene settings for real-world target retrieval experiments.

# D Additional Experimental Results

## D.1 Additional Results for Recognition Experiments

Additional examples of transparent object recognition results are shown in Figure 21 and Figure 22. The trends of these additional results are similar to the representative example in Section 5.1. For the test sets of the Tableware dataset in Figure 21, $T^2$SQNet succeeds in recognizing accurate 3D geometries of the transparent objects while also delivering instance information. GraspNeRF performs best among the baselines but predicts less accurate results than $T^2$SQNet. Although $T^2$SQNet has slightly lower performance on the Tableware dataset compared to TRansPose, it succeeds in predicting somewhat accurate instance-wise geometries, as shown in Figure 22.

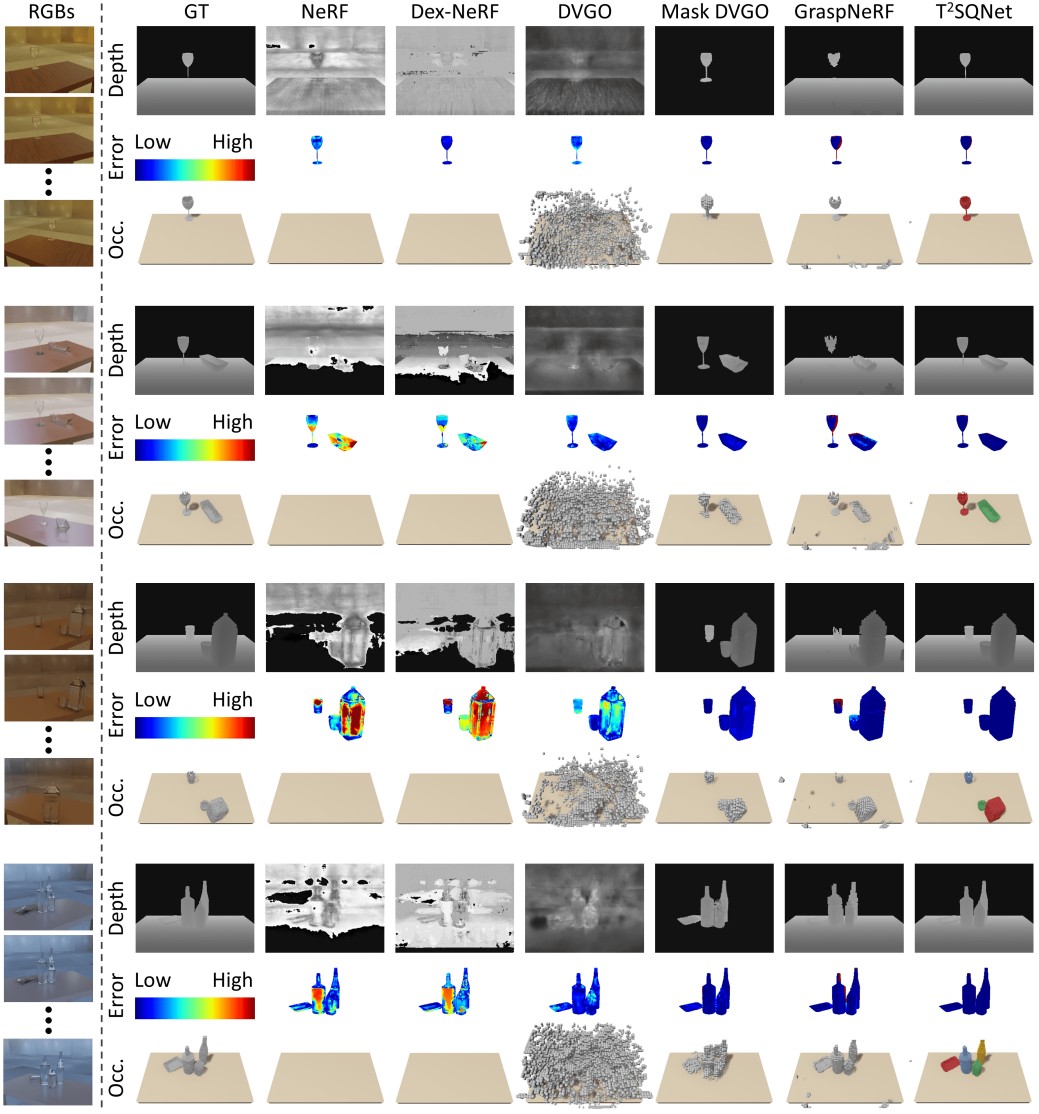

Figure 21: Recognition results from RGB images from test sets of Tableware dataset.

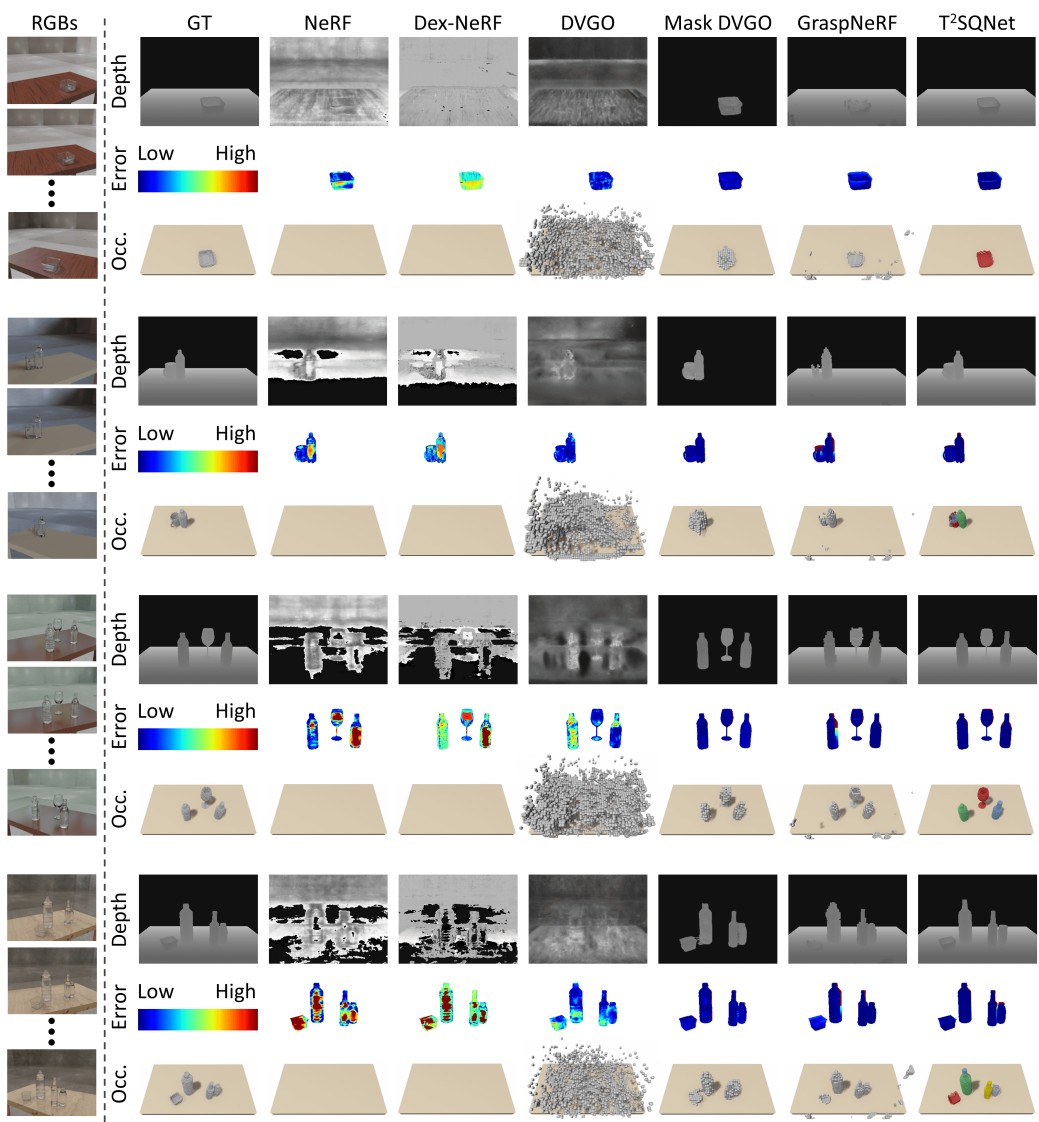

Figure 22: Recognition results from RGB images from TRansPose dataset.

**Additional Results for T²SQNet on TRansPose Objects.** We provide additional inference results of T²SQNet on TRansPose objects to (i) more closely compare the inference results of T²SQNet with the ground-truth surface shapes and (ii) assess how well T²SQNet generalizes to common transparent tableware objects. Figure 23 shows the ground-truth shapes of the transparent TRans-Pose objects alongside the inferred implicit surfaces from T²SQNet. Although capturing surface details such as the curvature of a water bottle is challenging due to the nature of superquadric surfaces, we can confirm that T²SQNet infers the overall shapes to a considerable extent. In particular, the predicted shapes appear accurate enough to perform manipulation tasks, including sequential declutter and target retrieval, which highlights the high performance of our method in real-world object manipulation scenarios.

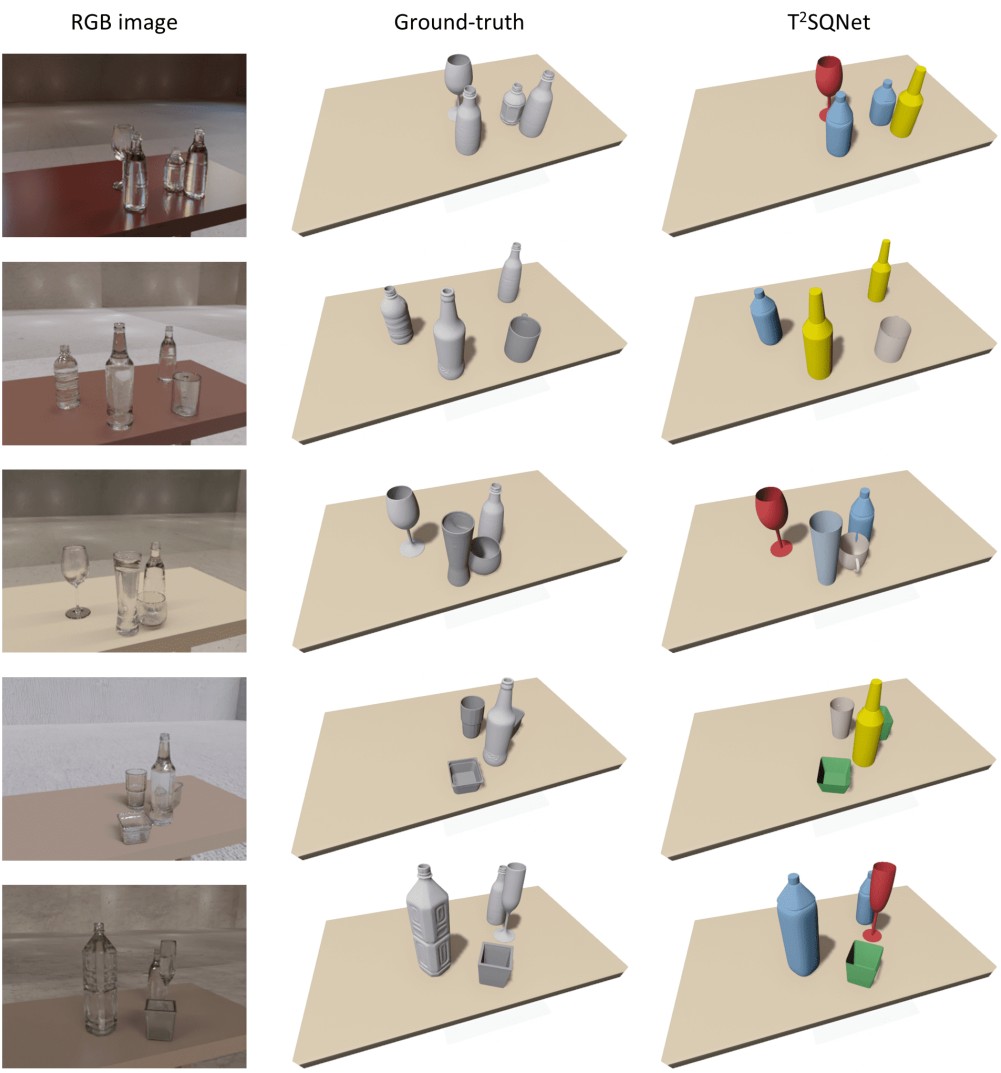

Figure 23: Recognition results of T²SQNet on TRansPose objects.

Additionally, we provide some failure cases of T$^2$SQNet on TRansPose objects, as shown in Figure 24. The first case (upper row of Figure 24) involves an error in the segmentation mask. In this instance, the long bowl was predicted to be short because the back part of the long bowl was not detected in the segmentation mask. The second case (middle row of Figure 24) involves an error in the bounding box predictor. The bounding box predictor fails to predict the bounding box of one bottle, so T$^2$SQNet does not predict any shape for the corresponding bottle. The third case is an interesting failure case for the bounding box predictor, where it gets confused between two object classes and predicts two shapes for one object. This suggests a potential improvement for the current T$^2$SQNet: Currently, when learning a bounding box predictor, bipartite matching is performed only within a class, and a loss function is provided. However, if we generalize this approach and perform bipartite matching between classes, we can avoid duplicate predictions. This is an interesting direction for our future work.

RGB image          Ground-truth          T$^2$SQNet

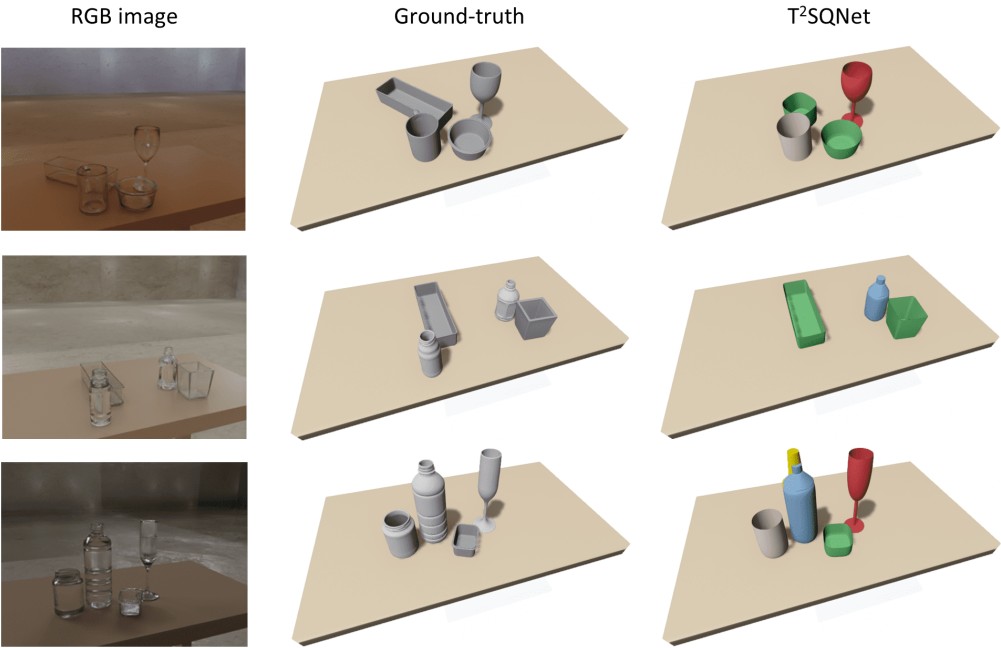

Figure 24: Failure cases in the recognition of T$^2$SQNet on TRansPose objects.

**Detailed Results and Analysis for Rendering-based Methods.** We describe the recognition results of rendering-based methods when provided with limited partial views and analyze the underlying reasons for their failure to predict the correct depth images. NeRF and Dex-NeRF are trained using seven partial camera view poses over a total of 200,000 iterations, and the results are shown in Figure 25. DVGO and Mask DVGO are also trained using the seven partial camera view poses. DVGO employs a two-stage training process consisting of a coarse stage with 5,000 iterations and a fine stage with 20,000 iterations. The results of DVGO and Mask DVGO are shown in Figure 26 and Figure 27.

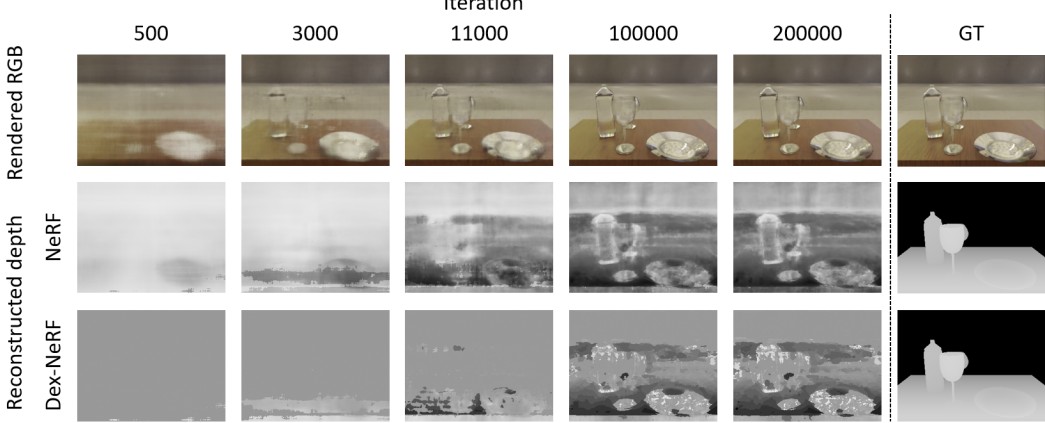

Figure 25: Results of RGB rendering and depth prediction for NeRF and Dex-NeRF trained on 7 partial views across training iterations.

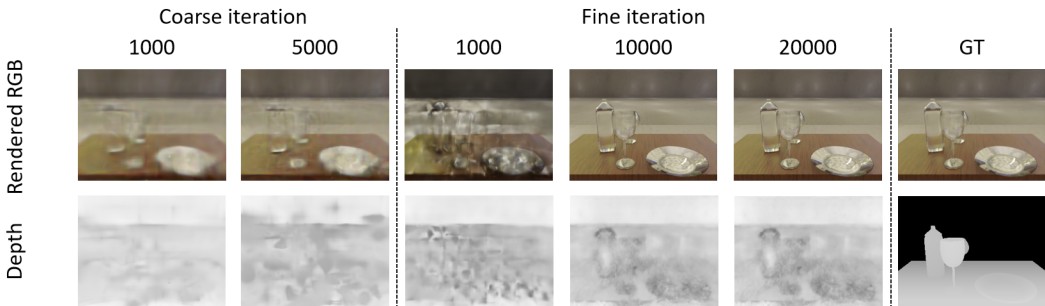

Figure 26: Results of RGB rendering and depth prediction for DVGO across training iterations.

As illustrated in the figures, the rendering-based methods, except for Mask DVGO, fail to predict accurate depth images even for the training views, despite successfully reconstructing RGB images for these views. Mask DVGO shows some success in predicting depth for the training views (e.g., the front view) but performs poorly in novel views (e.g., the side view).

The core issue is that the ultimate goals of NeRFs are to match the rendered images to the ground truth images. In other words, any radiance field that can generate the ground truth image can be a final solution, and there is no method to select which of these solutions accurately represents real-world geometry based solely on the given RGB information. Note that with sufficient view poses sampled from a hemisphere, the solution set can be expected to approximate real-world geometry; this has been demonstrated by Dex-NeRF and its subsequent study [3, 4]. However, the problem our work targets involves situations where only partial views are accessible, without the expectation of such extensive camera poses.

The figures above show how the problem of infinite solutions for generating the given RGB images becomes more severe with partial views. As shown in Figure 25, NeRF misinterprets the floor and

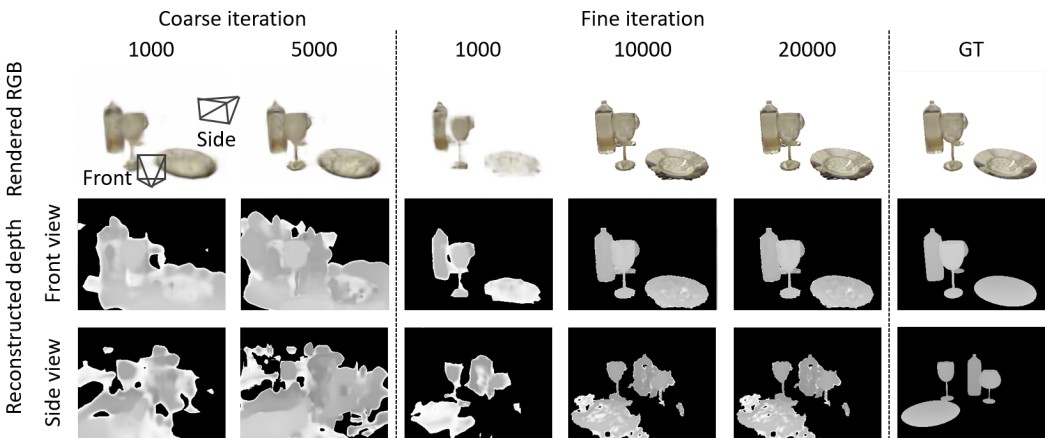

Figure 27: Results of RGB rendering and depth prediction for Mask DVGO across training iterations. The first two rows show the rendered results from the front view. The last row shows the rendered depth images from the side view, obtained by rotating the camera pose of front view 100 degrees around the z-axis of the world frame passing through the workspace center.

table surface depths as similar. Although the scene consists of a dark brown table placed on a light-colored floor, NeRF perceives the table as part of the floor with a darker brown area. The issue is more pronounced with DVGO, where most voxels are initially generated near the camera lens. The RGB values of these voxels are trained to match the real images at the lens proximity, resulting in very shallow depth values for all pixels, as observed in the Figure 26.

Mask DVGO, by utilizing mask information in addition to RGB, succeeds in predicting depth in training views. This success is attributed to the regularization term that forces points in rays outside the mask region to have zero opacity, effectively constraining opacity to exist only within the visual hull obtained from training mask images and view poses. While this visual hull functions well for depth observed near training view poses, it fails to accurately represent geometry in occluded or shadow regions, as these areas cannot be carved out. As shown in Figure 27, rendering side-view depth from Mask DVGO results in high opacity and depth in regions occluded from front views, indicating inaccurate geometry. One might argue that the accurate front view depth is sufficient for obtaining grasping poses using works like DexNet [31]. However, this is unsuitable for the target retrieval conducted in this paper, where reliable geometry information in occluded regions is crucial.

## D.2 Additional Results for Sequential Declutter Experiments

More examples of sequential decluttering with T²SQNet on shelves and tables are shown in Figure 28 and Figure 29, respectively. Generally, our method succeeds in sequentially grasping objects without re-recognition, while avoiding collisions with other objects and the environment based on the accurately predicted geometries of the objects. However, there are several failure cases: (i) a slightly incorrect shape leads to an unstable grasp pose, as shown in the third example of Figure 28, (ii) some objects are not recognized by T²SQNet, as shown in the first example of Figure 29, and (iii) an inverse kinematics solution does not exist, as shown in the third example of Figure 29. The first and second failure cases can be resolved through a more accurate recognition model, as described in the future works section (Section 5.3). The third failure case can be addressed by designing a more diverse 6-DoF grasp sampler.

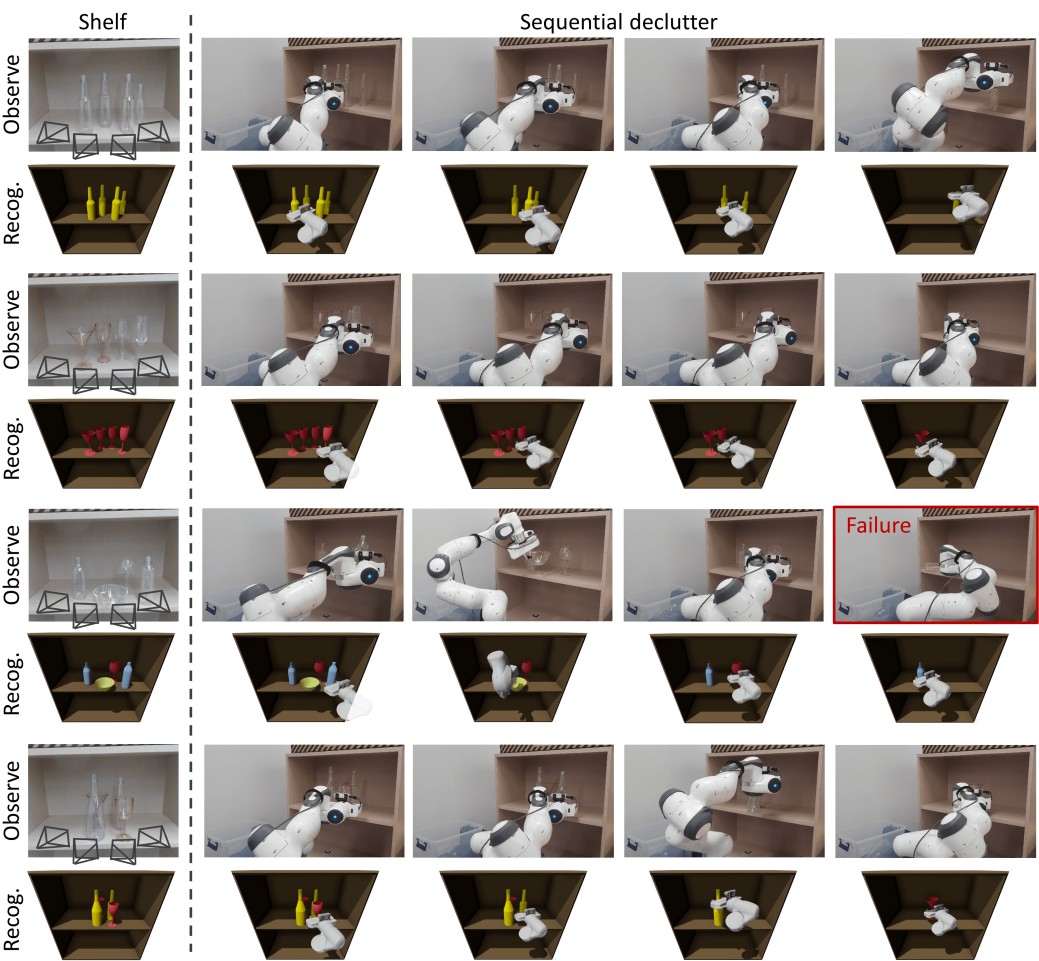

Figure 28: Examples of sequential declutter experiment results on shelves.

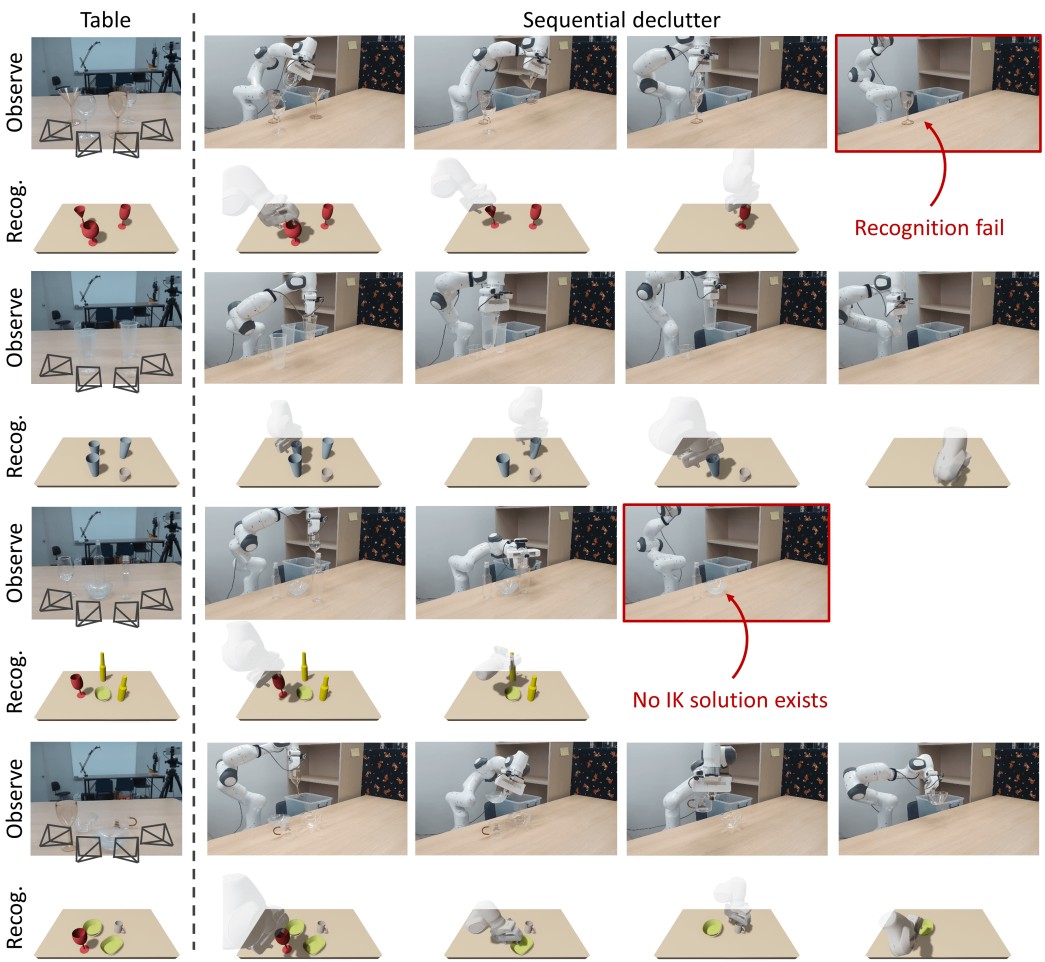

Figure 29: Examples of sequential declutter experiment results on tables.

## D.3 Additional Results for Target Retrieval Experiments

Additional results for target retrieval with $T^2$SQNet are shown in Figure 30. In the three examples above, $T^2$SQNet-based method successfully rearranges surrounding objects and retrieves target objects through appropriate pick-and-place actions. In the last example, $T^2$SQNet fails to recognize one wine glass; consequently, the robot performs an action of directly retrieving the target object, the mug, and as a result, it grasps both the wine glass and the mug together.

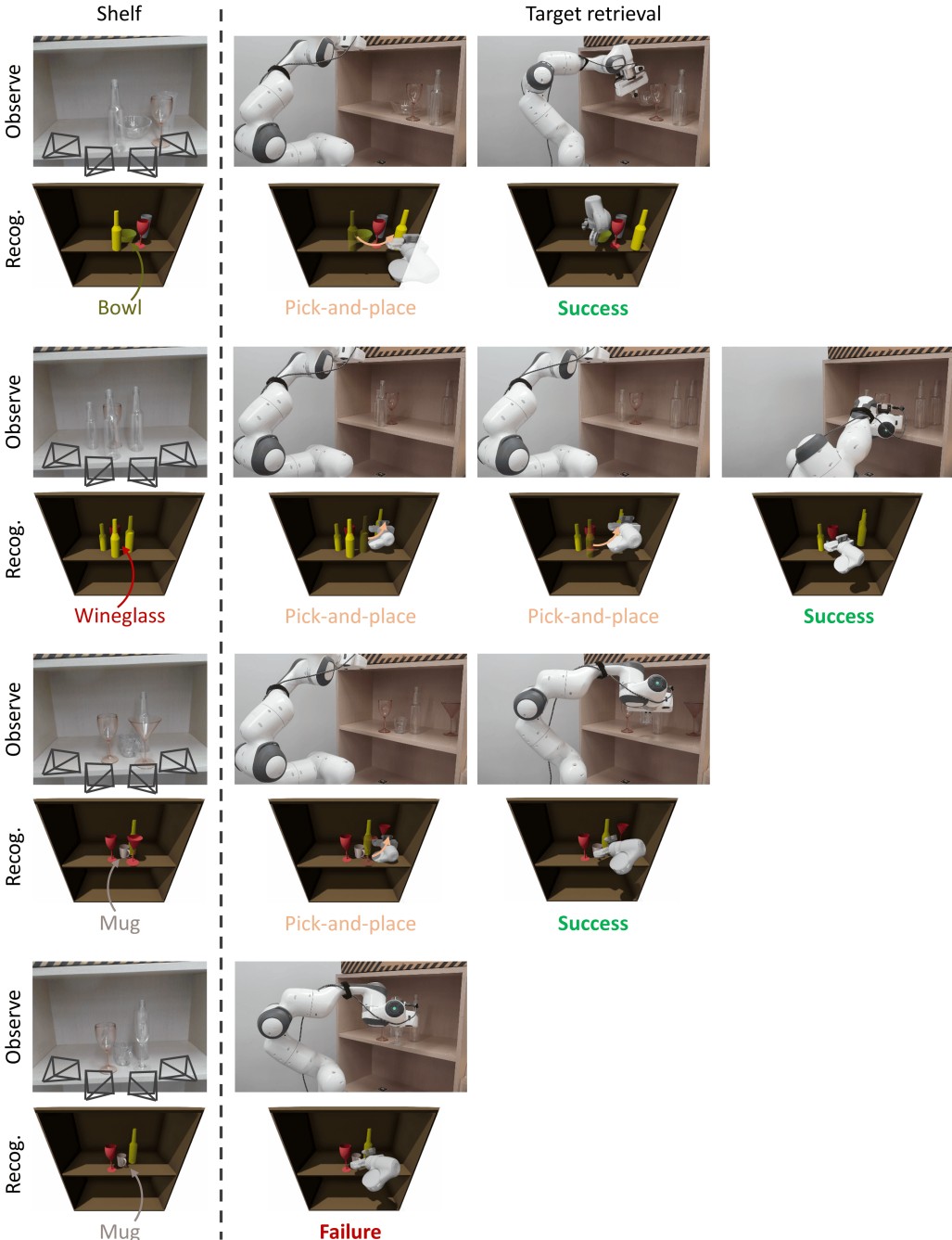

Figure 30: Examples of target retrieval experiment results on shelves.

### D.4 Comparison of T²SQNet with an End-to-End Method

To validate the effectiveness of our T²SQNet framework, which combines several separate modules, we developed and trained a simple end-to-end model for comparison. The end-to-end model structure is as follows: it utilizes our modified DETR3D structure (where queries are pre-assigned classes and a confidence estimation is incorporated) but replaces the bounding box predicting FCN with an FCN that predicts tableware parameters. Given that the dimension of tableware parameters varies by object class, separate FCNs are used for each class to predict the parameters. These class-specific FCNs take as input the queries assigned to the same class and output the corresponding tableware parameters. The learning loss for these parameters employs the same position regularizer, parameter regularizer, and chamfer loss as T²SQNet. In this end-to-end method, we set the weights for confidence loss, position regularizer, parameter regularizer, and chamfer loss to 1:1:1:0.1, respectively. The training results are shown in Figure 31. We observed that training this end-to-end model is considerably challenging, and we hypothesize the following reasons:

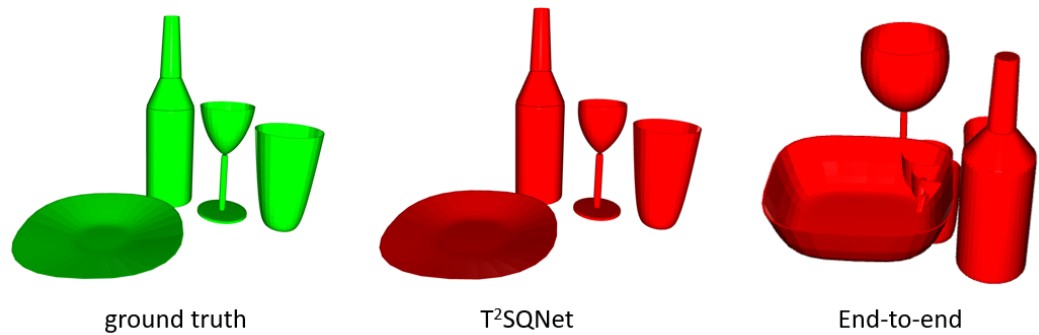

Figure 31: Visual comparison of T²SQNet output with an end-to-end method output on validation set

**Chamfer Loss Scale.** As the centers of two objects diverge, the chamfer loss scale increases quadratically. During the initial training phases, failing to align the object's position significantly inflates the chamfer loss scale relative to the confidence loss and position loss. Conversely, as objects come closer, the chamfer loss scale becomes much smaller than the confidence loss and other loss terms. This large variation in loss scale during training leads to inconsistent bipartite matching results, hindering significant training progress.

**Positional Constraints.** In T²SQNet, the object center position is constrained to remain within the bounding box, providing a structured framework for the output. However, the end-to-end method bypasses the bounding box prediction process, resulting in the lack of such positional constraints. Consequently, the predicted object can be located anywhere within the workspace during the initial training stages, making it difficult to resolve the aforementioned issues.

In conclusion, the large scale variation of the chamfer loss during training makes it challenging to balance the losses, leading to unstable bipartite matching results and ultimately hindering the learning process. While the end-to-end method theoretically streamlines the process, the practical challenges in training and the inherent issues in loss scaling and positional constraints highlight the advantages of our modular T²SQNet framework.

