# OpenReview forum: "T$^2$SQNet: A Recognition Model for Manipulating Partially Observed Transparent Tableware Objects"
_robot-learning.org/CoRL/2024/Conference — CoRL 2024_

### Official Review · Reviewer_hYHE · 2024-07-19
**review for T^2SQNet**

**Originality:** 3
**Technical Quality:** 4
**Clarity Of Presentation:** 4
**Potential Impact:** 3
**Recommendation:** 3
**Confidence:** 4

**Review:**

The task of identifying and manipulating clear tableware from partial view RGB images is indeed challenging due to the difficulty in obtaining reliable depth measurements of transparent objects. The Transparent Tableware SuperQuadric Network addresses this by using deformable superquadrics to create low-dimensional, instance-specific, and accurate 3D geometric representations of clear objects from partial views. Superquadrics are a parametric family of surfaces that can describe various shapes such as cubes, cylinders, and spheres using a continuous parameter space, making them a natural choice for geometric primitives due to their ability to model various shapes with few parameters.
Additionally, TablewareNet, a toolset of seven parameterized shapes based on these extended deformable superquadrics, allows for the creation of new datasets of tableware objects of various shapes and sizes. Experiments with T^2SQNet trained with TablewareNet have shown that T^2SQNet surpasses existing methods in identifying clear objects, making it highly effective for robotic applications such as decluttering and target retrieval.

The idea for using deformable superquadric to generate diverse transparent objects is novel compared to existing methods. Preview methods either use existing cad model or procedure generation to create rendered scenes. deformable superquadric allows for better diversity and more complex object shapes. But I think discuss the difference between TablewareNet and existing dataset with procedure generation is needed, for example SynTODD. And the ability to not only recover 3d bounding box or depth but the complete object shape is very help for robot manipulation task, which is shown in the paper. However, the novelty for the T^2SQNet architecture is questionable. Given that pretrained SAM, DETR3d and resnet3d are all published work, using them to form a pipeline is not novel in my opinion.

**Quality Of The Limitations Section:**

3

**Questions For Rebuttal:**

1. justify the novelty of T^2SQNet pipeline, what is authors contribution over SAM DETR3D and ResNet3D?
2. compare deformable superquadric object generation with procedure generation method used in SynToDD

**Robotics Focus:**

3

**Summary Of Paper:**

The task of identifying and manipulating clear tableware from partial view RGB images is indeed challenging due to the difficulty in obtaining reliable depth measurements of transparent objects. The Transparent Tableware SuperQuadric Network addresses this by using deformable superquadrics to create low-dimensional, instance-specific, and accurate 3D geometric representations of clear objects from partial views. Superquadrics are a parametric family of surfaces that can describe various shapes such as cubes, cylinders, and spheres using a continuous parameter space, making them a natural choice for geometric primitives due to their ability to model various shapes with few parameters. Additionally, TablewareNet, a toolset of seven parameterized shapes based on these extended deformable superquadrics, allows for the creation of new datasets of tableware objects of various shapes and sizes. Experiments with T^2SQNet trained with TablewareNet have shown that T^2SQNet surpasses existing methods in identifying clear objects, making it highly effective for robotic applications such as decluttering and target retrieval.

**Summary Of Recommendation:**

The task of identifying and manipulating clear tableware from partial view RGB images is indeed challenging due to the difficulty in obtaining reliable depth measurements of transparent objects. The Transparent Tableware SuperQuadric Network addresses this by using deformable superquadrics to create low-dimensional, instance-specific, and accurate 3D geometric representations of clear objects from partial views. Superquadrics are a parametric family of surfaces that can describe various shapes such as cubes, cylinders, and spheres using a continuous parameter space, making them a natural choice for geometric primitives due to their ability to model various shapes with few parameters. Additionally, TablewareNet, a toolset of seven parameterized shapes based on these extended deformable superquadrics, allows for the creation of new datasets of tableware objects of various shapes and sizes. Experiments with T^2SQNet trained with TablewareNet have shown that T^2SQNet surpasses existing methods in identifying clear objects, making it highly effective for robotic applications such as decluttering and target retrieval.  The idea for using deformable superquadric to generate diverse transparent objects is novel compared to existing methods. Preview methods either use existing cad model or procedure generation to create rendered scenes. deformable superquadric allows for better diversity and more complex object shapes. But I think discuss the difference between TablewareNet and existing dataset with procedure generation is needed, for example SynTODD. And the ability to not only recover 3d bounding box or depth but the complete object shape is very help for robot manipulation task, which is shown in the paper. However, the novelty for the T^2SQNet architecture is questionable. Given that pretrained SAM, DETR3d and resnet3d are all published work, using them to form a pipeline is not novel in my opinion.

---

### Official Review · Reviewer_p6K4 · 2024-07-20
**Weak Accept of paper on modeling and grasping transparent tableware objects.**

**Originality:** 4
**Technical Quality:** 4
**Clarity Of Presentation:** 2
**Potential Impact:** 3
**Recommendation:** 3
**Confidence:** 4

**Review:**

The paper is well-motivated in that categories of tableware typically share a large similarities in shape, and so compressing this shape to a smaller-parameter space makes sense.  However, the big question is how much geometric detail is needed in practice.  My sense is that for many robot tableware tasks, it is not necessary to recover the precise details of each object shape.
- This paper introduces an interesting extended class of deformed object shapes relevant to home tableware: TablewarNet.  The dataset can be a valuable source of data for those interested in transparent object recognition.
The recognition algorithm, T^2SQNet, seems to work for estimating object shape but the examples seem to be somewhat contrived in that the objects are within shelves and do not appear to exhibit reflections and specularities.
- The paper and detailed Appendix reads well, is thorough and clear, and has good figures to explain the method.  The video is also clear.
- There are ample baselines/comparisons to contextualize results against other relevant works.
- The experiments are thorough and clearly show the benefits of the proposed approach on the defined narrow set of tableware objects.
Weaknesses:
The paper seems to be overly dismissive of prior work with NeRF-based approaches, stating that they overfit and producing highly inaccurate geometries.  It would be extremely helpful to provide a detailed comparison to justify this claim, Table 1 and the very detailed Appendix are helpful but does not provide details on how and why previous methods fail.
- The method is highly constrained to transparent objects which can be represented by superquadrics, though one could argue this covers a large amount of everyday tableware. It would not work for complex objects like figurines or sculptures, or perhaps categories with more variation like free-form vases.
- It seems to take a fair amount of manual work to define a set of parameters for a category-specific superquadric, and is unclear if the system performance could scale to a large range of objects without significant effort (or if even with this effort, network performance could take advantage of the additional training data scale).

**Quality Of The Limitations Section:**

3

**Questions For Rebuttal:**

-The paper states that NeRF-based approaches “overfit and producing highly inaccurate geometries”.  It would be very helpful to provide MUCH more intuition and detail on where prior methods fail.  The examples in the paper and Appendix are very hard to see and it would be better to go though one example in much more detail.
Can the authors add inferences from the network of the output superquadrics, I only see occupancy visualized (except maybe figure 1). It would be useful to see the actual implicit surfaces themselves and compare against the ground truth.
- Details on the runtime of a single scene would be appreciated, for example time from when images have been captured to final output superquadrics.
Please address the concern above that the examples in the paper and video seem to be somewhat contrived in that the objects are within shelves and do not appear to exhibit reflections and specularities.
- Please clarify the details of the tableware parameter prediction. Because there can be a different number of tableware parameters per tableware category, does the Resnet3D predict also the category of object each is? Or is a resnet trained per-category (wineglass, bowl, etc), and the semantics from DETR3D are used to choose which resnet to use?
Please also say more about the limitation mentioned in the Limitations section, about the fixed set of 7 camera viewpoints.  Is this required for both training and testing?  Please show how the method performs if those 7 are not available.

**Robotics Focus:**

4

**Summary Of Paper:**

This paper studies reconstruction and manipulation of transparent tableware objects from multi-view images. It proposes an extension to the superquadrics implicit function representation for representing non-convex surfaces, dubbed superparabaloids, that includes deformation relevant to common tableware, then implements a set of tableware objects as a collection of these surfaces. The authors come up with a generation function for sampling each category of object, and use this to generate a space of test objects which they provide as a synthetic dataset. They then propose a soft voxel carving method for estimating the geometry of objects from an input set of multi-view masks, obtained from SAM. Given this voxel grid and 3D bounding box, they pass the grid through a 3D convnet to obtain a final set of category-specific super-quadric parameters, which is trained on the proposed synthetic dataset. Synthetic experiments evaluate the reconstruction ability of the system, and physical experiments evaluate its usefulness for multi-stage decluttering tasks involving planning trajectories to remove objects from a scene (shelf or table).

**Summary Of Recommendation:**

The paper and Appendix addresses an important problem in robotics.  Overall the material is well executed and thorough, so I would recommend weak acceptance. My reservations are mainly about the experiments and understanding precisely why this method works better than prior methods, and the technical motivation for the algorithm, because at its core the method seems relatively hard to scale and constrained to very specific categories of objects.

---

### Official Review · Reviewer_3VBo · 2024-07-20

**Originality:** 2
**Technical Quality:** 3
**Clarity Of Presentation:** 5
**Potential Impact:** 2
**Recommendation:** 2
**Confidence:** 4

**Review:**

Strength:

- The method proposed in this paper works pretty well on the reconstruction of transparent objects.
- The TablewareNet dataset itself is a good contribution to the research community.
- The real-robot experiments make the method more convincing.

Weakness:

- Not sure if the proposed method overfits the shape of the objects being reconstructed. Since the method includes learning-based segmentation and shape prediction, I assume that the method is trained on a subset of TablewareNet objects and tested on a held-out/unseen set of TablewareNet objects. It would be helpful to see the train/test object split to see how well this method can generalize.
- Relying on several Tableware Parameters to parameterize the objects is a weakness. What about other objects that cannot be easily parameterized? In this regard, the comparison in Figure 7 and Table 1 is not a valid comparison, because the method proposed in this paper relies on a very strong prior knowledge (shape parameterization) while other 3D reconstruction methods do not have such strong prior.
- It would be nice to see some quantitative evaluation of the manipulation (e.g. success rates, form closure etc) instead of several images plus a real-robot demo. A evaluation of the manipulation should be another core component of the method.

**Quality Of The Limitations Section:**

3

**Questions For Rebuttal:**

- How well can this method generalize to transparent objects that are never seen during training?
- The authors may want to compare their method to [A] in terms of 3D reconstruction quality.
- The authors can reply to other comments and issues raised in the "Weakness" part of my review above.

[A] "Through the Looking Glass: Neural 3D Reconstruction of Transparent Shapes", Zhengqin Li, Yu-Ying Yeh, Manmohan Chandraker, CVPR 2020

**Robotics Focus:**

4

**Summary Of Paper:**

This paper deals with the manipulation of partially transparent objects. The core contribution is on the perception part, where the authors proposed a framework to reconstruct the geometry and shape of the 3D transparent objects. There is also a synthetic dataset generated. Real robot experiments are shown.

**Summary Of Recommendation:**

Not sure if object shape parameterization is a good way to approach the problem. This severely limits the scope of application of this method. This is also the reason that the comparison against prior work might be unfair..

---

### Author Rebuttal · Authors · 2024-08-09

Thank you very much for your constructive feedback. In response to the many
constructive suggestions we have received, we have spent the past week
revising our manuscript accordingly. In an attempt to answer the reviewer
questions, and to better clarify and validate our contributions, we include
the new additional contents in the revised manuscript as follows:

* To validate our recognition model's effectiveness, we have added some additional experimental results in Appendix D.1 (Additional Results for Recognition Experiments).

* To highlight the effectiveness of our model in object manipulation, we have added details of manipulation techniques using deformable superquadrics in Appendix B.5 (Details for Geometry-Aware Object Manipulation).

* We have added a comparison between the TablewareNet dataset and existing datasets involving transparent objects in Appendix B.2 (Details for Tableware Dataset).

* We have supplemented some training details of our method in Appendix B.4 (Details for Model Architecture and Training Process).

* We have added details about the recognition experiments, including visualizations of the tableware objects used in the test and the runtime of our method, in Appendix C.1 (Additional Details for Recognition Experiments).

* We have modified Section 5.3 (Limitations and Future Directions) to describe the limitations of our approach more clearly.

* We have fixed some typos and expressions and clarified some statements.

Below we provide detailed responses to each of the reviewer comments.
When referencing any major changes and addition of new content to the
revised manuscript, we have indicated those passages ``like this``. The revised paper and appendix have been attached as a zip file.

---

### Decision · Program_Chairs · 2024-09-04

**Decision:**

Accept

**Comment:**

Thank you for your submission to CoRL 2024. The reviewers appreciated many aspects of the paper, particularly the real-robot experiments and high-quality reconstructions, good figures, videos, explanations, comparisons and baselines, and the novel idea of using super quadratic parameterization to generate a diverse set of objects. Some reviewers also pointed out that the tableware dataset makes a nice contribution on its own.

Reviewers raised concerns about the method's ability to generalize outside the training set and its reliance on object parameterization, which may limit its ability to work on complex objects in realistic settings. They also raised concerns about the novelty of the architecture and the dismissiveness of prior related work. A reviewer suggested that seeing it applied to a robot manipulation setting would strengthen the paper.

While reviewers appreciated the authors' responses, they were not sufficiently convinced to raise their recommendation. This suggests that there is still more work to do, and as such, we encourage the authors to continue improving the paper for the camera-ready version.